



**OMI Total Column Water Vapor Version 4  Validation and Applications**
Huiqun Wang[1], Amir Hossein Souri[1], Gonzalo Gonzalez Abad[1], Xiong Liu[1] and Kelly Chance[1]
[1]. Smithsonian Astrophysical Observatory, 60 Garden Street, Cambridge, Massachusetts 02138,
USA
*Correspondence to: Huiqun (Helen) Wang (hwang@cfa.harvard.edu)*
**Abstract**

9        Total Column Water Vapor (TCWV) is important for the weather and climate. TCWV is

derived from the OMI visible spectra using the Version 4 retrieval algorithm developed at the
Smithsonian Astrophysical Observatory. The algorithm uses a retrieval window between 432.0
and 466.5 nm and includes various updates. The retrieval window optimization results from the
trade-offs among competing factors. The OMI product is characterized by comparing against
commonly used reference datasets - GPS network data over land and SSMIS data over the
oceans. We examine how cloud fraction and cloud top pressure affect the comparisons. The
results lead us to recommend filtering OMI data with cloud fraction < 5 - 15% and cloud top
pressure > 750 mb or stricter criteria, in addition to the main data quality, fitting RMS and
TCWV range check. The mean of OMI-GPS is 0.85 mm with a standard deviation (σ) of 5.2
mm. Smaller differences between OMI and GPS (0.2 mm) occur when TCWV is within 10 – 20
mm. The bias is much smaller than the previous version. The mean of OMI-SSMIS is 1.2 – 1.9
mm (σ = 6.5 – 6.8 mm), with better agreement for January than for July. Smaller differences
between OMI and SSMIS (0.3 – 1.6 mm) occur when TCWV is within 10 – 30 mm. However,
the relative difference between OMI and the reference datasets is large when TCWV is less than
10 mm. As test applications of the Version 4 OMI TCWV over a range of spatial and temporal
scales, we find prominent signals of the patterns associated with El Niño and La Niña, the high
humidity associated with a corn sweat event and the strong moisture band of an Atmospheric
River (AR). A data assimilation experiment demonstrates that the OMI data can help improve
WRF's skill at simulating the structure and intensity of the AR and the precipitation at the AR
landfall.



## 1 Introduction

Water vapor is of profound importance for weather and climate. Through condensation, it forms clouds that modify albedo, affect radiation and interact with particulate matter. In addition, latent heat released from water vapor condensation can influence atmospheric energy budget and circulation. Water vapor is the most abundant greenhouse gas, accounting for ~50% of the greenhouse effect (Schmidt et al., 2010). Thus, monitoring the spatial and temporal distributions of water vapor is crucial for understanding water-vapor related processes.

Water vapor has been measured using a variety of in-situ and remote sensing techniques from the surface, air and space. Satellite data provide global perspective and are indispensable for constraining reanalysis products (Dee et al., 2011; Gelaro et al., 2017). The current satellite water vapor datasets are evaluated through the Global Energy and Water cycle Exchanges (GEWEX) Water Vapor Assessment program (Schroder et al., 2018). These datasets are derived from visible, near infrared (NIR), Infrared (IR), microwave and GPS measurements. Each dataset has its own characteristics. For example, microwave data are useful for both clear and cloudy conditions, but are best suited for non-precipitating ice-free oceans due to the complications associated with land surface emissivity; NIR data are best suited for the land, as the surface albedo is low over the oceans; IR data are available over all surface types, but are strongly influenced by clouds and less sensitive to the planetary boundary layer; visible data are sensitive to the boundary layer over both land and the oceans, but are complicated by uncertainties in clouds and aerosols (Wagner et al., 2013).

Total Column Water Vapor (TCWV, also called Integrated Water Vapor - IWV, or Precipitable Water Vapor - PWV) can be retrieved from the $7\nu$ water vapor vibrational polyad (around 442 nm) despite the band's weak absorption (Wagner et al., 2013). This made it possible to derive TCWV from instruments measuring in the blue wavelength range. Since water vapor is a weak absorber here, saturation of spectral lines is not of concern (Noël et al., 1999). Moreover, the similarity between the land and ocean surface albedo in the blue wavelength range suggests a roughly uniform sensitivity of the measurement over the globe (Wagner et al., 2013). However, weaker absorption tends to result in larger relative uncertainties, especially for low TCWV amount. As an example, for the Version 4 retrieval investigated in this paper, when TCWV is



greater than 10 mm, the medium fitting uncertainty is 10 – 15%, but for TCWV less than 10 mm,
it rises to 40 – 50%.

Using the visible spectra measured by the Ozone Monitoring Instrument (OMI), Wang et al.

(2014) retrieved Version 1 TCWV from 430 – 480 nm and publically released the data on the
Aura Validation Data Center (AVDC, https://avdc.gsfc.nasa.gov). Wang et al. (2016) found that
the Version 1 data generally agree with ground-based GPS data over land, but are significantly
lower than the microwave observations over the oceans. They found that using a narrower
retrieval window (427.7 – 465 nm) in Version 2 could improve the data over the oceans without
adversely affecting the results over land much. However, the Version 2 data were only generated
for a few test months and not released publically. An interim Version 3 OMI TCWV product is
available at AVDC. Compared with Version 2, Version 3 uses the latest reference spectra for
water vapor (Gordon et al., 2016) and liquid water (Mason et al., 2016), as well as the newest
cloud product (Veefkind et al., 2016). The Version 3 retrieval window (427.0 – 467.0 nm) is
adjusted from that for Version 2 within 2 nm on each end based on fitting uncertainty. However,
as discussed later, we find that the Version 3 data show much larger bias than the latest Version
4. Therefore, this paper focuses on Version 4 which will replace Version 3 on AVDC.        In this
paper, we present Version 4 OMI TCWV retrieval which incorporates a more vigorous
systematic optimization for the retrieval window and miscellaneous updates. We characterize the
performance of the Version 4 dataset by comparing with well-established references, such as the
GPS network data and SSMIS microwave observations. To provide practical information to
users of the new data, we investigate the influence of cloud fraction and cloud top pressure on
the comparisons. Based on the results, data filtering criteria is recommended. As an additional
check on the Version 4 product, we show test applications of the data to a range of spatial and
temporal scales, including El Niño / La Niña, a corn sweat event and an Atmospheric River (AR)
event. For the first time, a data assimilation experiment for the AR event examined demonstrates
that OMI TCWV data can provide useful constraint for weather prediction.
**2 Retrieval Algorithm**

OMI on board the AURA spacecraft is a UV/Visible imaging spectrometer (Levelt et al.,

2006). It has been making daily global observations at a nominal 13×24 km nadir resolution





around 1:30 PM since October 2004. The UV-Visible channel of OMI covers ~350-500 nm at a
spectral resolution of about 0.5 nm.

TCWV is derived from the OMI visible spectrum using a commonly used two-step approach.

First, the Slant Column Density (SCD, molecules/cm$^2$) is retrieved from a spectral fitting
algorithm. Then, the Vertical Column Density (VCD, molecules/cm$^2$) is calculated from the ratio
of SCD and Air Mass Factor (AMF) (Palmer et al., 2001). VCD can be converted to TCWV
using $10^{23}$ molecules/cm$^2$ = 29.89 mm. The details of the two-step procedure can be found in
Gonzalez Abad et al. (2015). The specifics of Version 4 is discussed below.

The Version 4 spectral fitting parameters are summarized in Table 1. In addition to water

vapor, we consider wavelength shift, under-sampling, closure polynomials (3$^{rd}$ order
multiplicative and additive), interfering molecules (O$_3$, NO$_2$, O$_4$, liquid water, C$_2$H$_2$O$_2$ and IO)
and Raman scattering (the Ring effect, vibrational Raman scattering of air and the water Ring
effect) in the non-linear least square fitting. In comparison with previous versions, Version 4 no
longer uses common mode (i.e. the mean fitting residual) in the fitting. It turns out that the
common mode for land is different than that for ocean (Wang et al., 2014), but previous
retrievals derive a common mode for each orbit swath using all the pixels in the low latitudes
which often includes both land and ocean scenes. Thus, the derived common mode depends on
the proportion of land versus ocean pixels of the spacecraft orbit and is not necessarily suitable
for all pixels. Statistics for Orbit 10423 shows that although the mean of SCD differs little
between the retrievals with and without common mode in the fitting (0.1 mm), the standard
deviation of SCD between them can be significant (1.7 mm). Most of the settings in Table 1 are
shared between Version 3 and 4, except that Version 3 uses Gordon et al. (2016) as the water
vapor reference spectrum, includes common mode, but does not consider vibrational Raman
scattering of air (Lampel et al., 2015).
**Table 1.** Parameters used in Version 4 spectral fitting for OMI total column water vapor.

| Wavelength shift | Solar reference spectrum | Dobber et al. (2008) |
|---|---|---|
| Target | H$_2$O | 288K, Rothman et al. (2009) |
| Interference molecules | O$_3$ | 228K, Brion et al. (1993) |
| | NO$_2$ | 220K, Vandaele et al. (1998) |
| | O$_4$ | 293K, Thalman and Volkamer (2013) |
| | Liquid water | Mason et al. (2016) |
| | C$_2$H$_2$O$_2$ | 296K, Volkamer et al. (2005) |





| | IO | 298K, Spietz et al. (2005) |
|---|---|---|
| Raman scattering | Ring effect | Chance and Spurr (1997) |
| | Water Ring | Chance and Spurr (1997) |
| | Air Vibrational Raman | Lampel et al. (2015) |
| Other | Additive polynomial | 3rd order |
| | Multiplicative polynomial | 3rd order |
| | Under-sampling | Chance et al. (2005) |


To optimize the retrieval window, we use OMI Orbit number 10426 (on July 1, 2006) as an
example to examine the effect of varying the starting and ending wavelengths around the $7\nu$
water vapor absorption band. The orbit swath contains $60\times1644$ ground pixels and covers parts
of Australia, the Pacific, China and other areas. We systematically adjust the starting wavelength
within 426.0-435.0 nm and the ending wavelength within 460.0-468.5 nm, both at 0.5 nm steps.

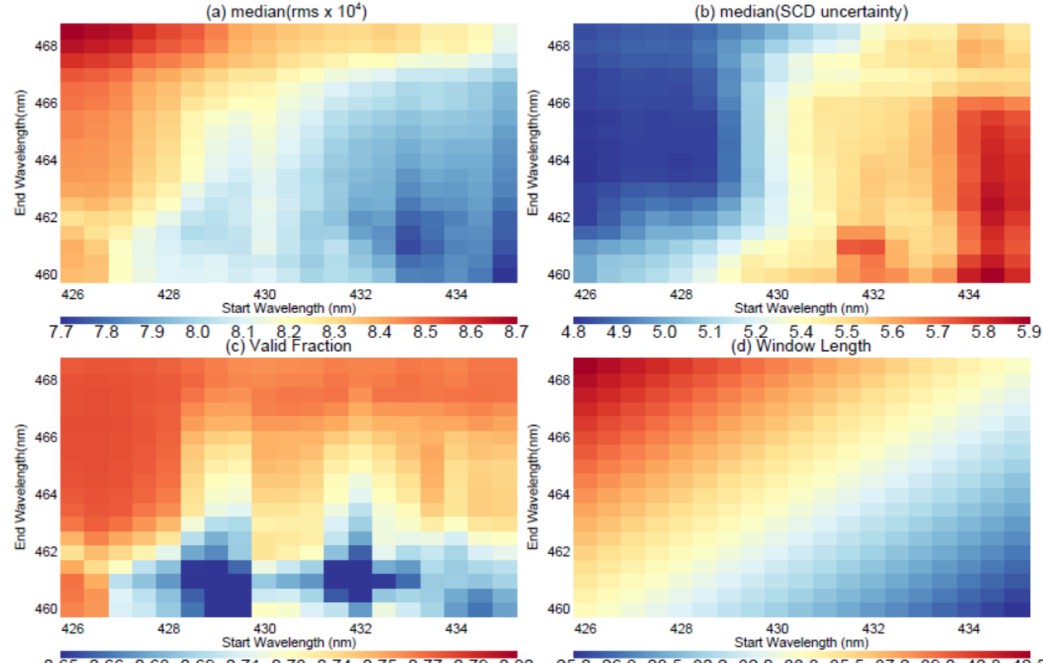


**Figure 1.** Sensitivity of the OMI TCWV retrieval to the start and end wavelengths (nm) of the
retrieval window. (a) Median of fitting RMS$\times10^{4}$; (b) median of water vapor SCD fitting
uncertainty in mm; (c) valid fraction; (d) retrieval window length in mm.
Previously, fitting window is based on fitting uncertainty. For Version 4, we consider four
factors (Figure 1). Figure 1a shows that the median of fitting RMS varies between $7.7\times10^{-4}$ and





$8.7 \times 10^{-4}$, and is smaller toward the lower right corner of the domain. Figure 1b shows that the
medium fitting uncertainty of water vapor SCD varies between 4.8 mm and 5.9 mm, and
decreases toward the upper left corner. Figure 1c shows that the fraction of valid retrievals for
the orbit varies between 0.59 and 0.78, and generally increases toward the upper part of the
domain. Valid retrievals here refer to those that pass the main data quality check (MDQFL = 0)
and have positive SCDs. The main data quality check ensures that the fitting has converged, the
SCD is $< 5 \times 10^{23}$ molecules/cm$^2$ and within $2\sigma$ of the fitting uncertainty. Figure 1d shows that the
length of the retrieval window varies between 25.0 nm and 42.5 nm, and increases toward the
upper left corner of the domain.
Ideally, we would like to have small fitting RMS to reduce the residual, a small fitting
uncertainty to reduce error, a large fraction of valid data to increase data volume and a long
retrieval window to include more information into the fitting. However, these criteria cannot be
met simultaneously. As a compromise, we select the wavelength interval between 432.0 nm and
466.5 nm as the retrieval window for Version 4. This leads to a median RMS of $8.1 \times 10^{-4}$, a
median uncertainty of 5.4 mm, a valid fraction of 0.75 and a window length of 34.5 nm.
Figure 2 shows that the median SCD varies between 34.6 mm and 37.6 mm (a 3 mm
difference corresponding to ~8% variation) and has a complex pattern within the domain. The
Version 4 retrieval window (432.0 – 466.5 nm) leads to a median SCD = 35.5 mm which is near
the beginning of the middle third of the SCD range. As will be shown in Section 3, the variation
of SCD in Figure 2 is quite large compared with the mean differences between OMI TCWV and
reference datasets.



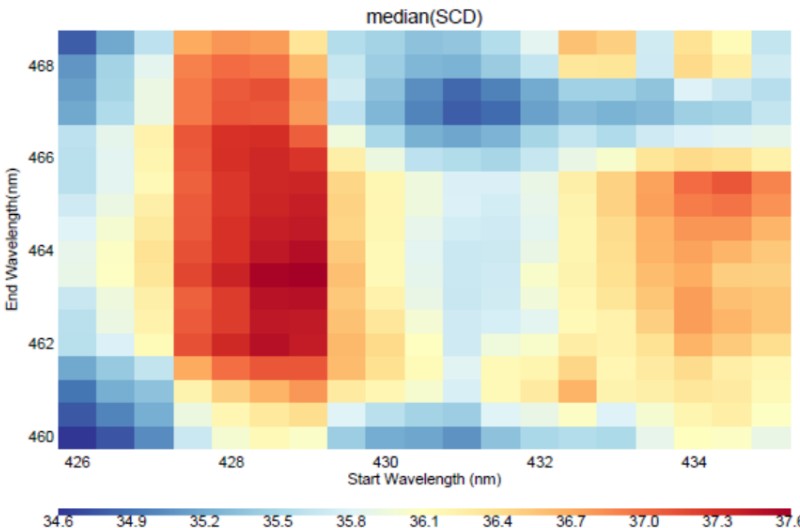

**Figure 2.** Sensitivity of the OMI water vapor median SCD (mm) to the start and end wavelengths (nm) of the retrieval window.

AMF is calculated by convolving scattering weights with the shape of water vapor vertical profile. The scattering weight is interpolated from the same look-up table as that used in Wang et al. (2016). The scene specific information used in the AMF calculation is listed in Table 2. Version 4 uses the 0.5°×0.667° monthly mean MERRA-2 water vapor profile (Gelaro et al., 2017) for the month and year corresponding to the retrieval, while previous versions used 2°×2.5° monthly mean of 2007 for all years. AMF is highly sensitive to clouds (Wang et al., 2014; Vasilkov et al., 2017). Version 4 uses the cloud information from Veefkind et al. (2016). The primary difference with the Acarreta et al. (2004) product used in Version 1 and 2 is in the cloud top pressure for cloud fraction < 0.3. In addition to the factors in Table 2, aerosol and surface bi-directional reflectance distribution function (BRDF) influence AMF (Lorente et al., 2017; Vasilkov et al., 2017), but have not been considered in the operational Version 4 yet.

**Table 2.** Parameters used in AMF calculation

| | |
|---|---|
| Solar Zenith Angle | OMI L1B data |
| View Zenith Angle | |
| Relative Azimuth Angle | |
| Surface Albedo | OMLER (Lambert equivalent reflectance) Kleipool, et al. (2008) |



| Cloud fraction | OMCLDO2 (derived from $O_2$-$O_2$) Veefkind et al. (2016) |
|---|---|
| Cloud top pressure | |
| Surface pressure | MERRA-2 monthly data (0.5°×0.667°), Gelaro et al. (2017) |
| Water vapor profile | |

## 3 Validation

To validate the Version 4 OMI TCWV data, we compare them against two commonly used reference datasets – a GPS network dataset for land and a microwave dataset for the oceans.

### 3.1 OMI and GPS over land

To assess the Version 4 OMI TCWV over land, we compare against the GPS network data downloaded from NCAR (rda.ucar.edu/datasets/ds721.1). The GPS data are composed of 2-hourly TCWV at International GNSS Service (IGS), SuomiNet and GEONET stations, and have an estimated error of < 1.5 mm (Wang et al., 2007; Ning et al., 2016). The subset of IGS-SuomiNet data over land for the whole year of 2006 is used in this paper.

OMI TCWV data are filtered using the following criteria. The stripes in Level 2 swaths due to systematic instrument error are removed using the SCD scaling procedure described in Wang et al. (2016). The pixels affected by row anomaly are filtered out (projects.knmi.nl/omi/research/product/rowanomaly-background.php), as well as unphysical (negative or extremely large) values. For "clear" sky comparison (Figure 3), we require radiative cloud fraction < 5% and cloud top pressure > 750 mb in addition to MDQFL = 0 and fitting RMS < 0.005.

To co-locate GPS and OMI data, we select the GPS data observed between 1100 LT and 1600LT. This local time range covers the OMI overpass time around 1330 LT. We average the qualified OMI data within 0.25° longitude × 0.25° latitude of the GPS station for each day. To minimize the influence of local topography (e.g., mountain peaks, river valleys), if a station's elevation is more than 500 m different than the mean elevation within the corresponding 0.25°×0.25° grid square, then it is excluded from the analysis. We consider the OMI and GPS data that are less than 75 mm. The co-locating procedure leads to 11,595 co-located data points distributed among 238 stations for 2006. Most of the selected stations are concentrated in North America and Europe. Fewer are scattered on other continents.





Figure 3 shows the comparison between co-located GPS and Version 4 OMI TCWV for
2006. The top panel shows the histogram of OMI-GPS. The 1.0-1.5 mm bin corresponds to the
peak of the histogram, i.e., the mode of the distribution. The mean (median) of OMI-GPS is 0.85
mm (0.84 mm), with a standard deviation of 5.2 mm.
In comparison, (Version 3 OMI - GPS) has a mean of 2.8 mm with a standard deviation of
5.5 mm. The bias of Version 3 is about three times as large as that of Version 4. This is attributed
to the much larger SCD in Version 3 (Supplementary Fig 1a), as the AMFs of Version 4 and
Version 3 closely follow the 1:1 line (Supplementary Fig 1b). Sensitivity tests show that the
larger Version 3 SCD is mostly due to the water vapor reference spectrum. If the water vapor
reference spectrum in Version 4 is replaced with that of Version 3 (Test 1), then the median SCD
increases by about 4.5 mm for Orbit 10423 (Supplementary Figure 1c). Modifying the retrieval
window for Version 3 cannot sufficiently reduce the retrieved SCD, therefore cannot make
significantly better agreement with the GPS reference data. However, the sensitivity test alone
cannot determine which water vapor reference spectrum is actually more accurate because the
fitting includes many other interference molecules (Table 1) whose reference spectra may also
contain errors within the retrieval window. As Version 4 shows better performance, this paper
focuses on characterizing Version 4 and providing useful information to users of the data.
The bottom panel of Figure 3 shows the joint distribution of the co-located data. The count of
each bin is normalized by the maximum of all bins. About 46% of the data have TCWV < 10
mm, 83% have TCWV < 20 mm and 95% have TCWV < 30 mm. There is a general linear
correlation between GPS and OMI data, with a correlation coefficient of r = 0.805. The
regression line (OMI = 3.10 + 0.82 × GPS) has a significant positive intercept and a slope that is
less than one. This indicates a positive bias for small TCWV and a negative bias for large
TCWV. Indeed, as indicated in the bottom panel, the mean of OMI-GPS for each 10 mm GPS
TCWV bin decreases from 2.3 mm for TCWV = 0 – 10 mm to -2.9 mm for TCWV = 40 – 50
mm, though the fraction of data for TCWV > 40 mm is < 1%. The corresponding standard
deviation (σ) increases from 4.0 mm to 9.2 mm. The minimum bias of 0.2 mm occurs for the
TCWV = 10 – 20 mm bin. Since there are more data points for TCWV = 0 – 10 mm than for
TCWV = 10 – 20 mm, the peak in the top panel of Figure 3 lies between 0.2 mm and 2.3 mm.





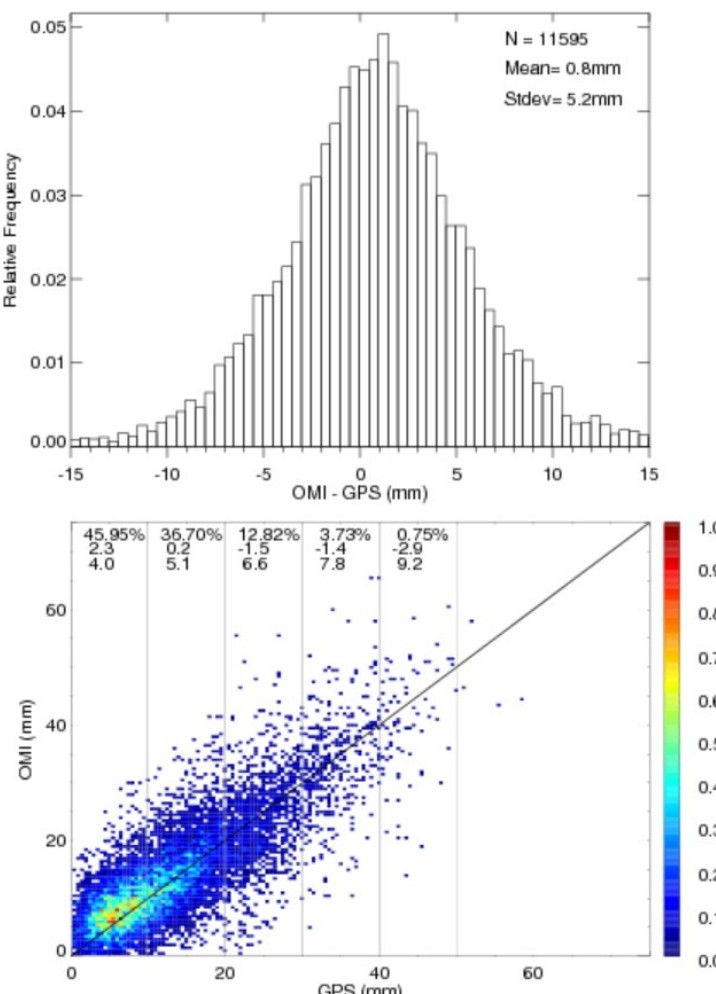

**Figure 3.** Comparison between co-located GPS and OMI TCWV (mm) for all days in 2006. The data filtering criteria include cloud fraction < 5%, cloud top pressure > 750 mb, and others discussed in the text. (Top) Relative frequency of occurrence for OMI-GPS (mm). The total number of data pairs, the mean and standard deviation of OMI-GPS (mm) are indicated in the upper right corner. (Bottom) Normalized joint distribution of GPS versus OMI TCWV (mm). At the top of the panel, the three lines of text indicate the percentage of data points (top), the mean of OMI-GPS in mm (middle), and the standard deviation of OMI-GPS in mm (bottom) for each 10 mm GPS TCWV, respectively. The 1:1 is overplotted for reference.





The OMI TCWV retrieval is highly sensitive to clouds (Wang et al., 2014). Thus, in Figure
4, we examine the effect of OMI radiative cloud fraction threshold (f) (Gonzalez Abad et al.,
2015) on the comparison while keeping other data filtering criteria the same as those for Figure 3
(i.e., cloud fraction < f, cloud top pressure < 750 mb, MDQFL = 0 and fitting RMS < 0.005). The
number of co-located data pairs (N) increases with f, such that N more than doubles between f =
0.05 to f = 0.55. The mean of OMI-GPS increases from 0.85 mm to 1.7 mm as f increases from
0.05 to 0.55. The standard deviation of OMI-GPS increases by ~11% from f = 0.05 to f ≥ 0.45.
The linear correlation coefficient (r) increases rapidly from r = 0.805 at f = 0.05 to r = 0.855 at f
= 0.15, then levels off near r = 0.86 for larger cloud fraction thresholds. Therefore, f = 0.05 leads
to the lowest overall bias and scatter of the co-located data, but f = 0.15 leads to a ~50% increase
in the number of co-located data pairs and the largest improvement in the GPS versus OMI
correlation coefficient. Hence, cloud fraction thresholds of f = 0.05 – 0.15 seems a reasonable
choice for filtering OMI TCWV.

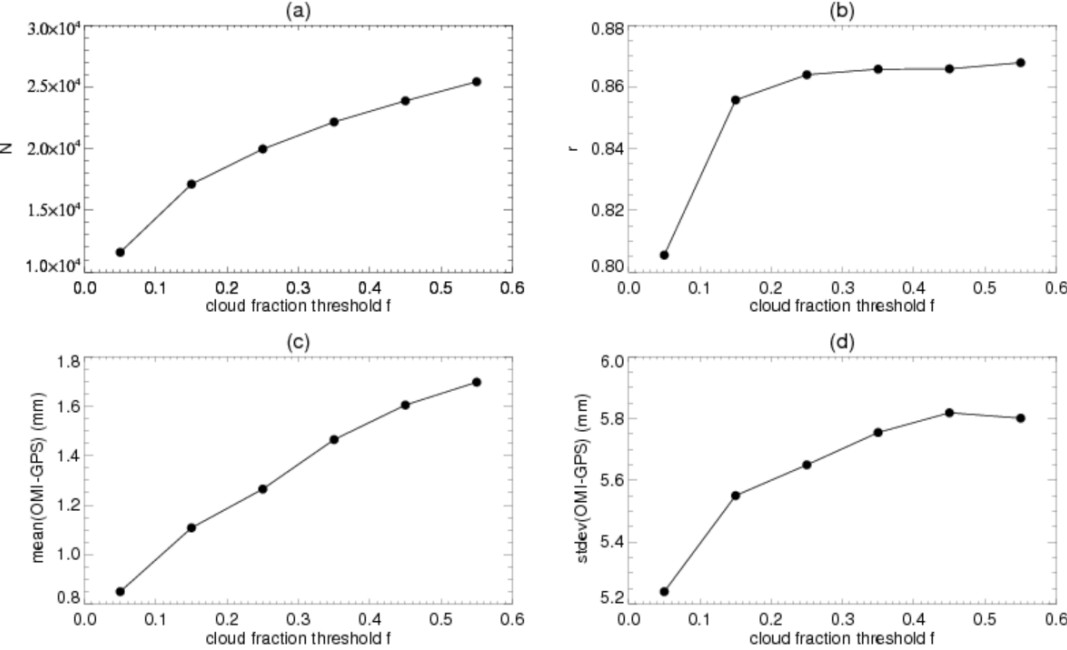


**Figure 4.** Dependence of various statistical parameters on the radiative cloud fraction threshold
(f) used for filtering OMI data. Other filtering criteria remain the same as those for Figure 3. The
parameters are (a) number of co-located OMI and GPS data pairs; (b) linear correlation



coefficient between OMI and GPS TCWV; (c) mean of OMI-GPS in mm; (d) standard deviation
of OMI-GPS in mm. Results are for 2006.

To further characterize the effect of cloud fraction threshold on the comparison, in Figure 5

we examine the mean and standard deviation (σ) of OMI-GPS for each 10 mm GPS TCWV
interval. The results are derived from the same sets of co-located GPS and OMI data as those
used in Figure 4. The filled symbols in Figure 5 are for the cases where the number of GPS and
OMI data pairs within the corresponding TCWV interval is > 1% of the total number of data
pairs within 0 – 60 mm, and the open symbols are for < 1%. Since the filled symbols represent
better statistics, we will focus on them below.

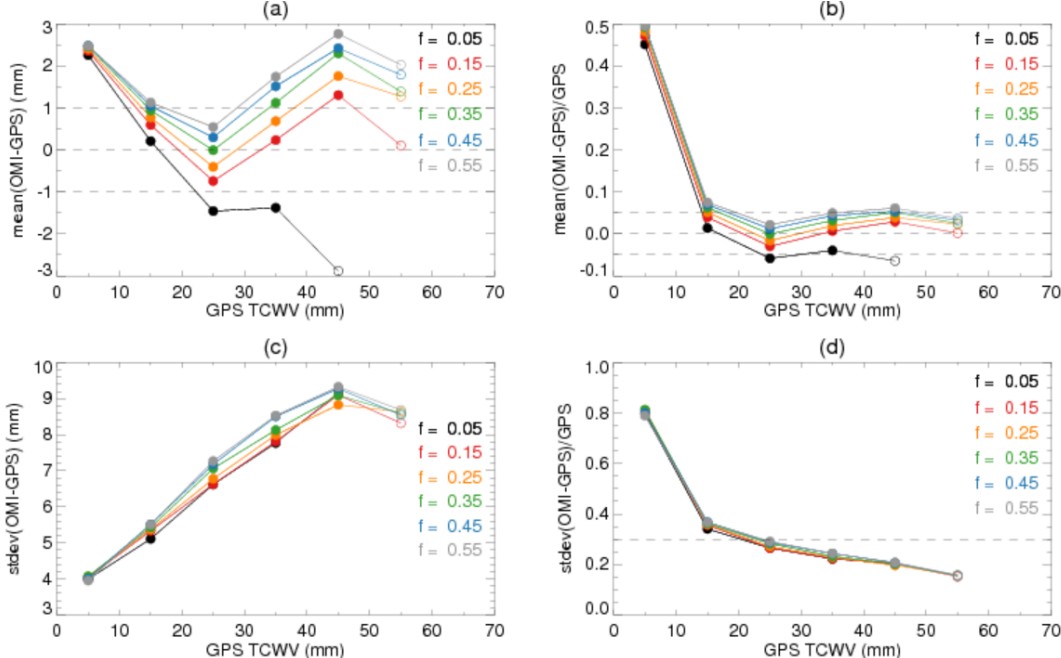


**Figure 5.** Statistical parameters for each 10 mm GPS TCWV interval. Curves with different
colors are for different radiative cloud fraction thresholds f. Other OMI filtering criteria remain
the same as those for Figure 3. Symbols are filled if the fraction of data pairs within the TCWV
interval is > 1% of all the available data pairs, and are open otherwise. The parameters are (a)





mean of OMI-GPS in mm, (b) relative bias = (OMI-GPS)/GPS, (c) standard deviation σ of OMI-GPS in mm and (d) relative scatter = σ/GPS. Results are for all days in 2006. Dashed lines are meant to facilitate visualization.

Figure 5a shows that the means of OMI-GPS vary between ±3 mm following "V"-shaped curves whose minima occur in the TCWV = 20 – 30 mm interval. The curves shift upward with increasing cloud fraction thresholds, suggesting that OMI cloudy TCWV is larger than OMI clear TCWV in general. The f = 0.15 and f = 0.25 curves show the best performance as they lie within 1 mm of zero for 10<TCWV<40 mm, while other curves come within 1 mm of zero in narrower TCWV ranges. Figure 5b shows the relative bias which is defined as mean (OMI-GPS)/GPS. The relative biases decrease sharply from 50% to ~5% as GPS TCWV increases from 0 – 10 mm to 10 – 20 mm, and generally stay less than ~5% for larger TCWV values. Figure 5c shows that σ ranges from 4 mm to 9.5 mm and increases with TCWV. In most cases, higher cloud fraction thresholds correspond to larger σ values. Figure 5d shows that the relative scatter (σ/TCWV) decreases with TCWV, with the sharpest decrease from ~0.8 to ~0.3 occurring between TCWV = 0 – 10 mm and TCWV = 10 – 20 mm. In short, Version 4 OMI agrees with GPS within 1 mm for 10<TCWV<40 mm when f = 0.15 – 0.25 is used; when f = 0.05 is used, the bias and scatter are the smallest for 10<TCWV<20 mm; but, OMI TCWV is too high and has large scatter for TCWV < 10 mm, as expected from the weak absorption of water vapor in the blue spectral range.

**3.2 OMI and microwave over ocean**

To evaluate Version 4 OMI TCWV over the oceans, we compare against the microwave TCWV data from the SSMIS (Special Sensor Microwave Imager/Sounder) instrument on board the Defense Meteorological Satellite Program (DMSP)'s F16 satellite. The SSMIS data are derived by the Remote Sensing Systems using their Version 7 algorithm (www.remss.com) and have a retrieval accuracy of better than 1 mm (Wentz, 1997; Mears et al., 2015). In this paper, we use the daily 0.25°×0.25° SSMIS data for January and July 2006 and filter out the pixels affected by rain and cloud liquid water. Diedrich et al. (2016) found that the diurnal cycle in TCWV is generally within 1% to 5% of the daily mean, with a minimum between 0600 LT and 1000 LT and a maximum between 1600 LT and 2000 LT, though larger diurnal cycle exist for





special cases. To reduce the influence of the diurnal cycle, we average the SSMIS data for the
ascending and descending orbits of F16 (~2000 LT and 0800 LT in 2006).

We generate daily 0.25°×0.25° Level 3 OMI TCWV from the de-striped Level 2 OMI

swaths, with the requirement that MDQFL = 0, fitting RMS < 0.005, 0<TCWV<90 mm, cloud
fraction < 0.05, and cloud top pressure > 750 mb. There are typically 15 Level 2 swaths per day.
The gridding program uses a tessellation method that weighs the contribution of a Level 2 data
point by its area within the Level 3 grid square and its spectrum fitting uncertainty. The filtered
daily Level 3 SSMIS and OMI data are compared for each month. We find 928,426 and 721,669
co-located data pairs for January and July 2006, respectively.

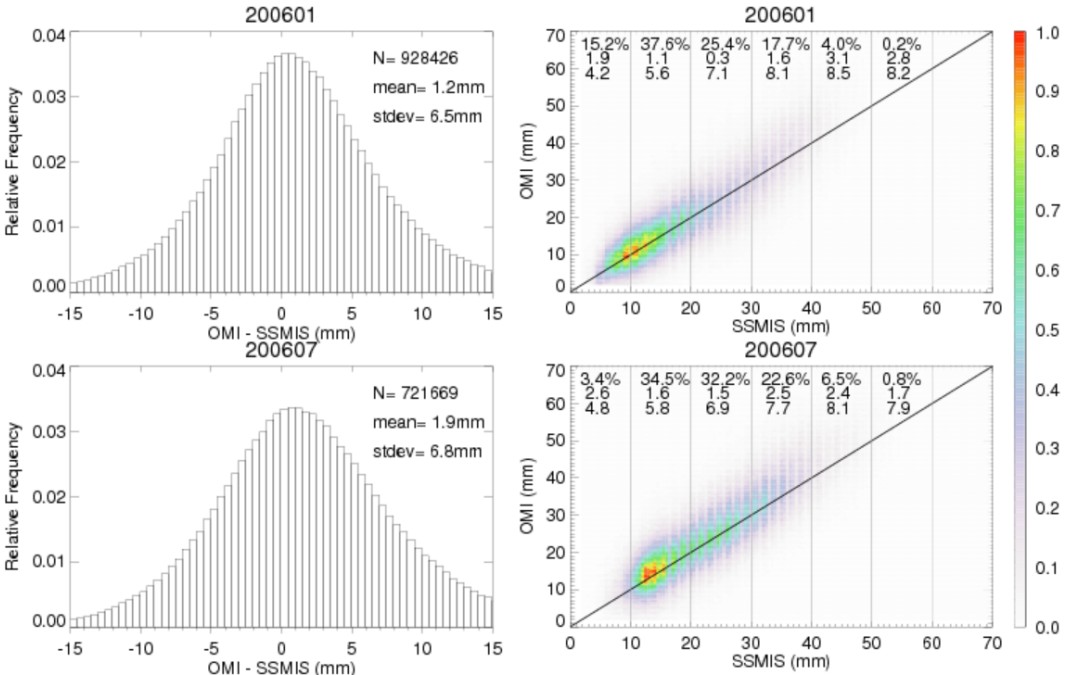


**Figure 6.** Comparisons between Version 4 OMI and SSMIS over the oceans for (top) January
2006 and (bottom) July 2006. Panels in the left column show the relative frequency of
occurrence (i.e., number of points within each bin / total number of points) of OMI-SSMIS
(mm). The total number of data pairs (N), mean and standard deviation of the distribution are
indicated in the upper right corners. Panels in the right column show the normalized joint
distribution of SSMIS versus OMI TCWV (mm). The 1:1 line is overplotted for reference. At the





top of each right panel, the three lines of text correspond to the percentage of data pairs (top), the
mean (middle) and the standard deviation (bottom) of OMI-SSMIS (mm) for each 10 mm
SSMIS TCWV bin indicated by the gray vertical lines.

The left column of Figure 6 shows the histogram distribution of Version 4 OMI-SSMIS for
January and July 2006. For July, the mean of OMI-SSMIS is 1.9 mm with a standard deviation
of 6.8 mm; for January, the corresponding values decrease to 1.2 mm and 6.5 mm, respectively.
This suggests a slightly better agreement for January than for July. In comparison with the OMI-
GPS over land (Section 3.1 Figure 3), the OMI-SSMIS over the oceans has somewhat larger bias
and standard deviation. However, as TCWV over the oceans are generally larger than that over
land (compare Figure 6 with Figure 3), the relative bias and scatter are actually similar.
The right column of Figure 6 shows the normalized joint distribution of SSMIS versus OMI
for January and July 2006. The correlation coefficients are r = 0.85 and 0.83 for January and
July, respectively. The mean of OMI-SSMIS for each 10 mm TCWV interval shows that OMI is
higher than SSMIS by 0.3 – 3.1 mm in January and by 1.5 – 2.6 mm in July. For both months,
the smallest absolute difference between OMI and SSMIS occurs for TCWV = 20 – 30 mm, and
the next smallest one occurs for TCWV = 10 – 20 mm. The standard deviation of OMI-GPS
increases from about 4 mm for TCWV = 0 – 10 mm to about 8 mm for TCWV > 40 mm. Thus,
OMI data compare well with SSMIS data for TCWV in the 10 – 30 mm range.
Table 3 shows the effect of the OMI radiative cloud fraction threshold (f) on the comparison
between SSMIS and Version 4 OMI TCWV. As before, the comparisons are performed using
daily filtered Level 3 data for July 2006. For SSMIS, we keep cloudy pixels except when they
are affected by rain; For OMI, we require MDQFL = 0, RMS < 0.005, cloud top pressure > 750
mb and cloud fraction < f when running the gridding program. Results show that OMI is higher
than SSMIS by 0.91 – 3.35 mm. The closest agreement in terms of the mean and standard
deviation of OMI-GPS occurs when f = 0.05, in which case, the regression line is OMI =
1.12+0.99×SSMIS. The number of SSMIS and OMI data pairs more than doubles between f =
0.05 and f = 0.15, and the linear correlation coefficient increases from 0.84 to 0.86. For larger
cloud fraction thresholds, although there are more data pairs, the correlation coefficients do not
improve, and the means and standard deviations increase. Therefore, for OMI TCWV over the





oceans, we recommend using cloud fraction threshold f in the 0.05 – 0.15 range, in combination
with the other usual data filtering criteria.
**Table 3.** Effect of cloud fraction threshold on the comparison between SSMIS and Version 4
OMI TCWV for July 2006.

| OMI cloud fraction threshold f | Number of data pairs | Mean(OMI-SSMIS) (mm) | Stdev(OMI-SSMIS) (mm) | Correlation coefficient r | Regression line |
|---|---|---|---|---|---|
| 0.05 | 1411842 | 0.91 | 7.04 | 0.84 | OMI=1.12+0.99*SSMIS |
| 0.15 | 3424330 | 2.37 | 7.57 | 0.86 | OMI=1.17+1.04*SSMIS |
| 0.25 | 4578487 | 2.93 | 7.75 | 0.86 | OMI=1.55+1.05*SSMIS |
| 0.35 | 5391356 | 3.21 | 7.79 | 0.86 | OMI=1.65+1.06*SSMIS |
| 0.45 | 6009664 | 3.35 | 7.77 | 0.86 | OMI=1.65+1.07*SSMIS |


Lowering the value for cloud top pressure threshold also leads to larger OMI TCWV and
therefore larger bias and scatter. For example, when cloud fraction < f = 0.05 and cloud top
pressure > 300 mb are used to filter OMI data for July 2006, the mean and standard deviation of
OMI-SSMIS become 2.6 mm and 7.5 mm, respectively. These values are approximately between
those for f = 0.15 and f = 0.25 when cloud top pressure > 750 mb is used (Table 3), and they are
larger than those shown in Figure 6. Relaxing the filtering criteria for both cloud fraction and
cloud top pressure will lead to larger bias and scatter, and is therefore not recommended. As an
example, for cloud fraction < 0.15 and cloud top pressure > 300 mb, the mean (standard
deviation) of OMI-GPS increases to 3.9 mm (7.9 mm) for July 2006.
**4 Application**
**4.1 El Niño / La Niña**
In Figure 7, we examine the signals associated with El Niño and La Niña in Version 4 OMI
TCWV. Panel (a) shows the Multivariate ENSO Index (MEI) from NOAA (Wolter and Timlin,
1998) (https://www.esrl.noaa.gov/psd/enso/mei/). Positive (negative) values correspond to El
Niño (La Niña) conditions. We examine the changes in TCWV for July 2010 (MEI = -1.103, La
Niña) and July 2015 (MEI = 1.981, El Niño) in the bottom row. Although these events are strong
within the OMI record (from 2005 to the present), they are mild in comparison with the extrema.
Between 1950 and 2018, the maximum MEI is 3.008 (in March 1983) and the minimum MEI is -
2.247 (in June 1955).




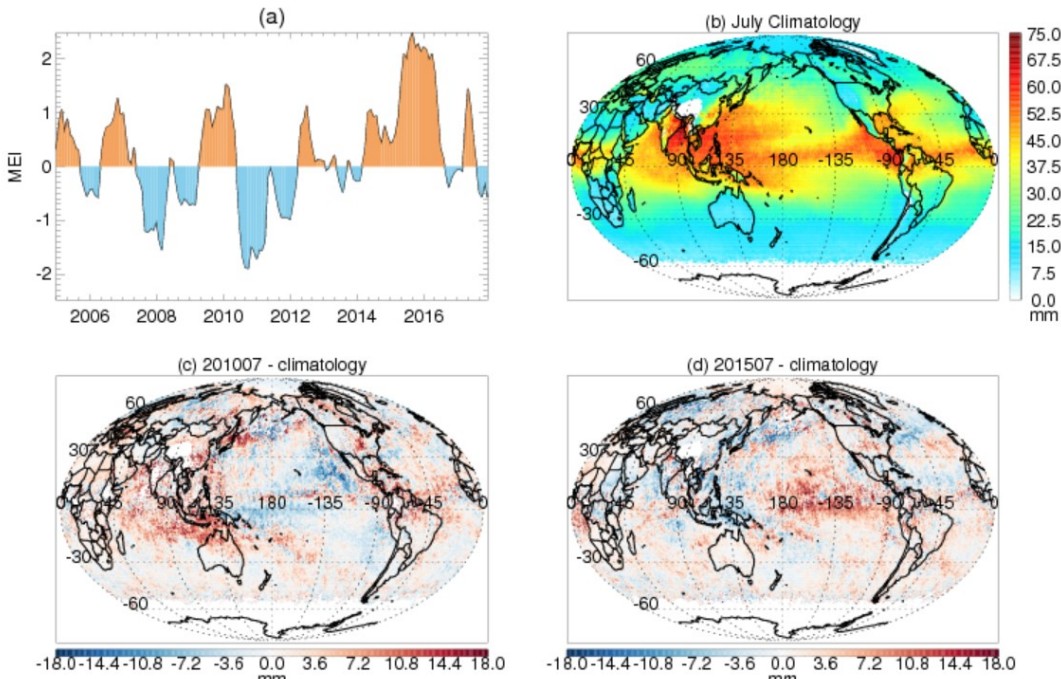


**Figure 7.** Top row: (a) Multivariate ENSO Index; (b) TCWV (mm) climatology for July derived from Version 4 OMI data between 2005 and 2015. Bottom row: TCWV anomaly (mm) with respect to the climatology for (c) July 2010 and (d) July 2015.


To examine the changes in OMI TCWV under different conditions, we first generate the monthly Level 3 (0.5°×0.5°) OMI TCWV for each July between 2005 and 2015 using the method described in Section 3.2 (with a cloud fraction threshold of 0.15 and a cloud top pressure threshold of 750 mb). Then, using the same data filtering criteria, we derive a climatology for July using all the Level 2 July data between 2005 and 2015 (Figure 7b). Finally, we plot the monthly deviations from the climatology (mm) for July 2010 and July 2015 in Figure 7cd.

The TCWV anomalies exhibit large-scale patterns. The pattern for July 2015 largely opposes that for July 2010. Particularly, in July 2015 under El Niño conditions, TCWV increases in the equatorial central and eastern Pacific and deceases in the Indonesia region; While in July 2010 under La Niña conditions, TCWV deceases in the tropical eastern Pacific and equatorial western



Pacific and increases in Indonesia and the Indian Ocean. The overall patterns largely conform to
the results derived from the Hamburg Ocean Atmosphere Parameters and Fluxes from Satellite
Data (HOAPS) data (Shi et al., 2018). The HOAPS climatology is derived from a longer time
series (1998-2014), which may be among the reasons for the differences in details between the
results.

**4.2 Corn Sweat**

"Corn sweat" refers to a hot and humid condition associated with heat waves which results in
large evapotranspiration rate in the Midwestern United States where cropland is often the
dominant land usage type. Besides evaporation, transpiration by plants, such as corn, draws
water from the soil to the atmosphere, enhancing the humidity and increasing the heat index. A
corn sweat made news in the US from July 18th to July 22nd of 2016. This event is examined in
Figure 8 using Version 4 OMI TCWV.
Figure 8 (a) and (b) show the Level 3 (0.25°×0.25°) OMI TCWV for July 17th - July 23rd (7-
day) and June 1st – August 31st (JJA) in 2016, respectively. The 7-day period covers the corn
sweat event. The Level 3 data are derived using the same data filtering criteria as those used for
Figure 7. The difference (a)-(b) shown in Figure 8(c) indicates the anomaly associated with the
corn sweat event relative to JJA mean. High TCWV is observed for the 7-day period from the
Gulf coast to the Midwestern US. Besides the Gulf region, the largest TCWV enhancements (of
up to 18+ mm) occur in parts of Iowa (IA), Missouri (MO), Illinois (IL) and Indiana (IN).



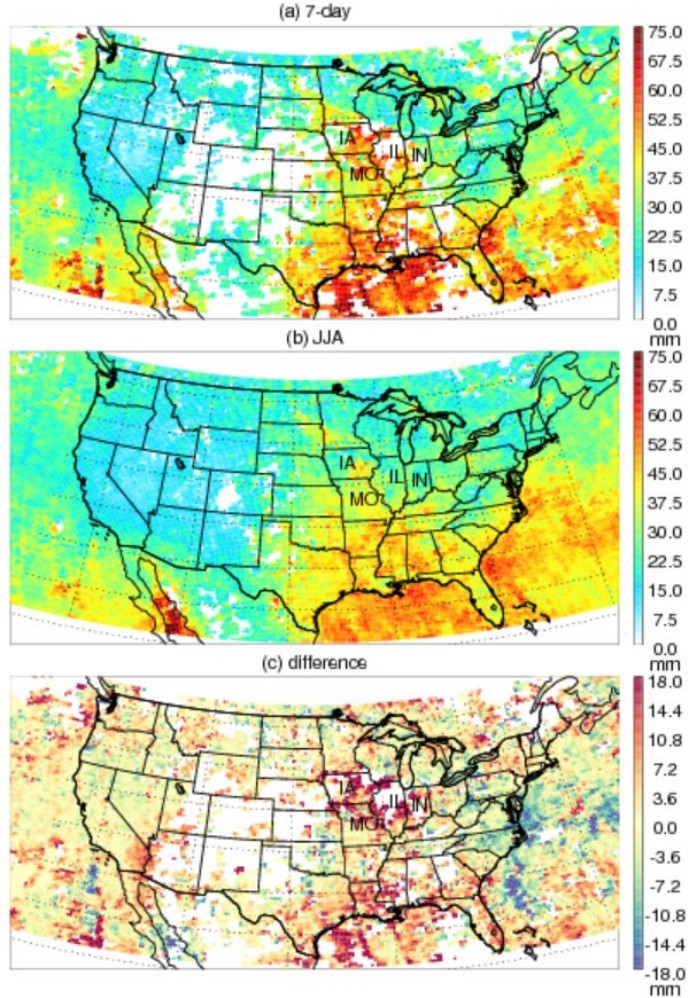


**Figure 8.** Level 3 (0.25°×0.25°) OMI TCWV (mm) generated using the Level 2 data during (a)
July 18 - July 23, 2016 and (b) June 1 - August 31, 2016. (c) The difference (a) - (b) in mm. The
abbreviations for the states most affected by the corn sweat event are indicated.


To assess the significance of evapotranspiration for the Midwestern US during the corn sweat
event, we carried out a sensitivity study using the Weather Research and Forecasting (WRF)
model v3.9.1 (Skamarock et al., 2008). The model was run on a 36-km parent domain and a 12-
km nested domain, covering the relevant areas of the US. The physics parameterizations
included the WRF Single-Moment (WSM) 6-Class Microphysics (Hong and Lim, 2006), the



Kain-Fritsch (KF) subgrid cumulus parameterization (Kain, 2004), the Yonsei University (YSU)
planetary boundary layer scheme (Hong et al., 2006), the Noah Land-Surface Model (Ek et al.,
2003; Chen and Dudhia, 2001), and the Rapid Radiative Transfer Model (RRTM). Horizontal
turbulent diffusion was based on the standard Smagorinsky first-order closure. The initial and
lateral boundary conditions were from the 3-hourly NARR reanalysis at 32-km resolution. To
reduce the uncertainty associated with lateral boundary condition of the nested domain, we
nudged the model values in the parent domain toward the reanalysis, but left the interior of the
nested domain running freely.

To diagnose the contribution of evapotranspiration, the model was run from July 19$^{th}$ to July

22$^{nd}$ of 2016 with and without evapotranspiration (calculated in the Noah LSM model). The
results for July 21$^{st}$ are shown in Figure 9. TCWV is generally lower in the run without
evapotranspiration (No ET). The difference between the runs suggests that evapotranspiration
contributes about 15 – 25% of the TCWV in the Midwestern US during the July 2016 corn sweat
event. A detailed study incorporating the OMI TCWV with the WRF model will be carried out in
future work.

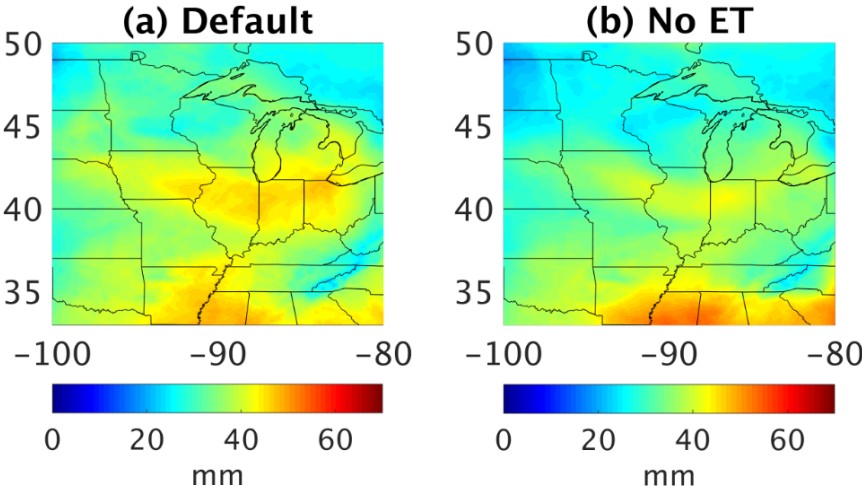


**Figure 9.** WRF simulations of TCWV (mm) for Midwestern US on 07/21/2016 for the run (a)
with and (b) without evapotranspiration.

**4.3 Atmospheric River (AR)**



ARs are narrow elongated bands with high TCWV in the atmosphere. With flow rates similar
to those of large rivers, ARs are highly important in the global hydrological cycle (Zhu and
Newell, 1998). Land-falling ARs can lead to heavy orographic precipitation that affects areas
such as the west coast of North America and Europe (Gimeno et al., 2014).

**4.3.1 An Intense AR**

The extreme AR of November $6^{th} – 7^{th}$, 2006 brought devastating flood to the Pacific
Northwest – the region in western North America bounded by the Pacific to the west and the
Cascade mountain range to the east. This AR is observed in SSM/I (Special Sensor
Microwave/Imager) TCWV data as a narrow band of high water vapor  stretching northeastward
from the moist rich equatorial central Pacific to the Pacific Northwest (Neiman et al., 2008).
Such an AR is usually nicknamed as a "Pineapple Express" by weather forecasters (Lackmann
and Gyakum, 1999). The vertical cross sections of specific humidity observed by COSMIC
(Constellation Observing System for Meteorology, Ionosphere, and Climate) show that the AR is
concentrated between the surface and 700 mb near the leading edge of a polar front which slopes
northwestward from the surface toward the tropopause (Neiman et al., 2008). The AR is
associated with a low level jet in the warm conveyor belt of an extra-tropical cyclone that
develops along the polar front. In the meanwhile, the GOES-11 6.7μm brightness temperature
image (for upper tropospheric water vapor) shows a curvilinear dark stripe that is parallel to and
west of the AR (Neiman et al., 2008). The dark stripe indicates subsidence of dry air from above.
It is consistent with the COSMIC potential temperature observations of stratospheric air
intrusion, signaling an upper-tropospheric jet stream (Neiman et al., 2008).

**4.3.2 The AR in OMI observation**

The signature of this AR is captured in Version 4 OMI TCWV data. The left column of
Figure 10 shows the Level 3 OMI TCWV and its anomaly on November $6^{th}$, 2006. The Level 3
data are generated following the same procedure as that used for Figure 7. Although many pixels
are missing because of the cloud filtering (cloud top pressure > 750 mb, cloud fraction < 0.15)
and other criteria, the leading edge of the AR is noticeable as an elongated band of high TCWV
(15+ mm above the climatology) extending from Hawaii to Northern California (indicated by
arrows in Figure 7bc). The position of the AR in OMI TCWV agrees well with that in SSM/I
observation (Neiman et al., 2008).


The right column of Figure 10 shows the Level 3 OMI ozone mixing ratio interpolated to 200
mb and its anomaly. The OMI ozone data are retrieved using the SAO ozone profile algorithm
(Liu et al., 2010; Huang et al., 2017, 2018). The climatology is derived by averaging all monthly
Level 3 data for November from 2004 to 2017. The global distribution of ozone at 200 mb shows
low mixing ratio in the low latitudes and high mixing ratio in the high latitudes, opposite to the
global distribution of TCWV. The ozone anomaly shows a curvilinear band that is parallel to the
AR (in the left column), but is located further to the west. This feature indicates intrusion of
ozone rich stratospheric air along the polar front, and is consistent with the dark stripe in the
upper tropospheric water vapor image obtained by GOES-11 (Neiman et al., 2008).

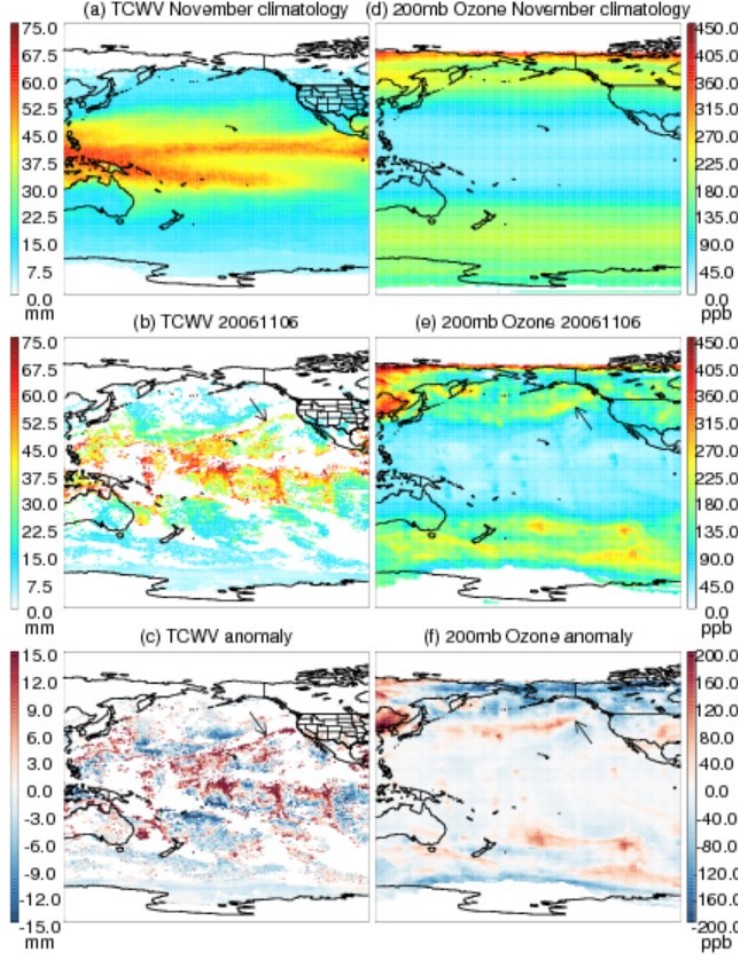




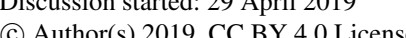

**Figure 10.** The Level 3 (top row) climatology, (middle row) data on November 6[th], 2016 and
(bottom row) anomaly on November 6[th], 2016 with respect to the climatology of (left column)
Version 4 OMI TCWV (mm, 0.5°×0.5°) and (right column) OMI ozone mixing ratio (ppb,
1°×1°) interpolated to 200 mb.

### 4.3.3 OMI Data Assimilation for the AR

To evaluate the potential of OMI water vapor data to improve numerical weather forecasts,
we conducted a data assimilation experiment from November 2[nd] to November 8[th] of 2016 using
WRF v3.9.1 and Version 4 OMI TCWV. The model was configured with a 27-km (290×270
surface grid points with 51 vertical levels), a 9-km (586×586×51 points) and a 3-km
(541×526×51) nested domains in a Lambert projection over the relevant portion of the Pacific
and North America (Figure 11 top left). The domains are designed for the November 6 AR event
and its associated precipitation at landfall. The model has the same physics parameterizations as
those used in Section 4.2 except that a more sophisticated double-moment microphysics scheme
is used in the 3-km nest for quantifying precipitation. The initial and boundary conditions for the
27-km domain were from the 1°×1° NCEP FNL reanalysis. One-way nesting is used for the
inner domains. To evaluate the model's skill at simulating the AR and the contribution of OMI
TCWV to the quality of the simulation, we did not nudge the run towards the reanalysis, nor
assimilate the observed sea surface temperature within the computational domains.
The OMI TCWV is assimilated into the model using analytical optimal estimation (Rodgers,
2000). This method minimizes the cost function $J(\boldsymbol{x}) = (\mathbf{y} - H\boldsymbol{x})^T \mathbf{E}^{-1}(\mathbf{y} - H\boldsymbol{x}) +$
$(\boldsymbol{x} - \boldsymbol{x}^b)^T \mathbf{B}^{-1}(\boldsymbol{x} - \boldsymbol{x}^b)$, where $x$ is the true TCWV, $x^b$ is the a priori TCWV (from the model), $y$
is the observed TCWV, $H$ represents the model Jacobian, $\mathbf{B}$ and $\mathbf{E}$ are the error covariance
matrices of the a priori and observation. $\mathbf{B}$ is estimated using the 12-hour and 24-hour forecasts
using the National Meteorological Center method (Parrish and Derber, 1992). $\mathbf{E}$ is based on the
fitting uncertainties of OMI data.
The a posteriori analysis ($\hat{\boldsymbol{x}}$) can be obtained from $\hat{\boldsymbol{x}} = \boldsymbol{x}^b + \mathbf{K}(\mathbf{y} - H\mathbf{x})$, where $\mathbf{K} =$
$\mathbf{B}H^T(H\mathbf{B}H^T + W^{-1}\mathbf{E})^{-1}$ is the Kalman gain, $W = \frac{(R^2 - r^2)}{(R^2 + r^2)}$ is the Cressman function to weigh the
observations based on their Euclidian distance $r$ to the model grids, and $R$ is the influence radius



of the observations. We simply assume $R$ to be $1^o$, $0.5^o$ and $0.25^o$ for the 27-km, 9-km and 3-km
domain to get a quick look at the results in this paper, and leave more vigorous quantification of
$R$ to future work.  The a posteriori TCWV is solved hourly when OMI data are available and is
used to initialize the next simulation window.
During the assimilation, we adjust the OMI data using the AMF calculated with the modeled
water vapor profile ($OMI^{adjusted}_{satellite} = \frac{OMI_{satellite} \times AMF_{satellite}}{AMF_{model}}$). This can reduce the observational
error associated with using the monthly mean water vapor profile in the operational OMI
product. The standard deviation of the difference between $AMF_{satellite}$ and $AMF_{model}$ is about 20%.

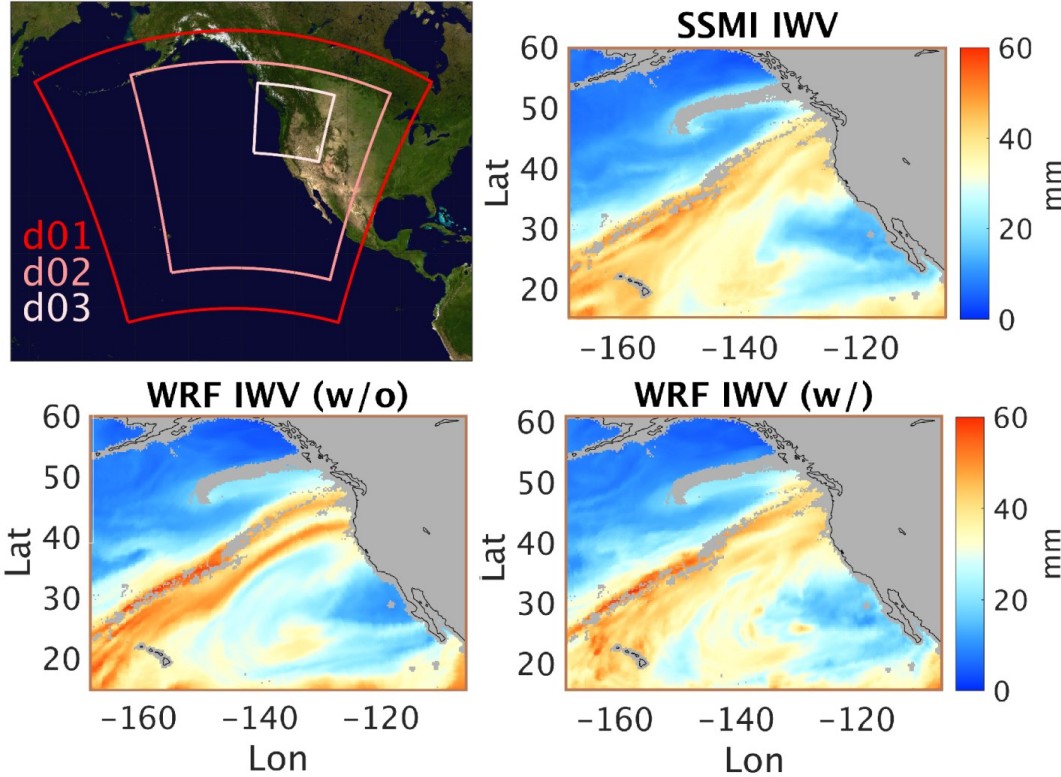


**Figure 11.** Top left: WRF model domain configuration for the November 2016 AR event. Top
right: TCWV observed by SSM/I on November 6[th], 2016. Bottom row: TCWV simulated by
WRF on November 6[th], 2016 (left) without and (right) with OMI data assimilation.





Figure 11 shows the zoomed-in views of the AR on November 6[th], 2016. The TCWV
independently observed by SSM/I is shown in the upper right panel. The lower left and lower
right panels show the model results without and with OMI TCWV assimilation. The model
without assimilation shows an AR that is split into two parallel filaments making landfall at
separate locations on the west coast of US, where the TCWV is too high compared to the SSM/I
observation, especially for the southern filament. This has significant impact on the precipitation
(Figure 12). After assimilating OMI TCWV, the modeled TCWV agrees much better with the
SSM/I observation. The overall shape and magnitude of the AR are significantly improved.
The location and intensity of precipitation over land are crucial for local flood control and
water management, and are closely related to the shape and strength of AR at landfall. The 24-
hour accumulated precipitation on November 6 in the 3-km domain is examined in Figure 12.
The model output is upscale to 0.25°×0.25° to match the resolution of the TRMM (Tropical
Rainfall Measuring Mission) observation product. The model without OMI data assimilation
erroneously produces rainfall over the Oregon - California border (box A) as a result of the error
in the simulated AR structure (Figure 11). This artifact was removed after using OMI data,
showing better agreement with the corresponding TRMM rainfall observation.

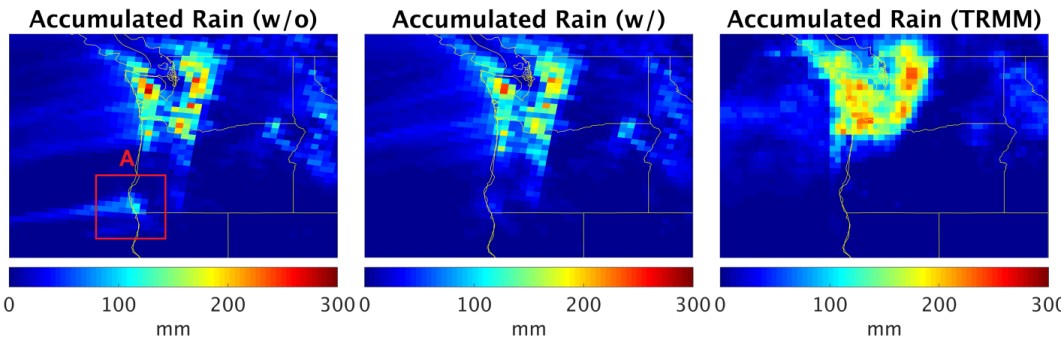


**Figure 12.** The simulated rainfall accumulated from 0000 UTC to 2300 UTC (in mm) on
November 6, 2006 for the model (left) without and (middle) with OMI TCWV assimilation. The
rightmost panel show the accumulated rainfall observed by TRMM for the same time period.
Note that the 3-km model result is coarsened to match the resolution of the TRMM product.
Box A highlights the erroneously simulated precipitation in the run without OMI data
assimilation.






**5 Summary and Conclusion**

The Version 4 retrieval algorithm for OMI Total Column Water Vapor (TCWV) is presented
in this paper. The algorithm follows the usual two-step approach where Slant Column Density
(SCD) is derived from spectral fitting and Vertical Column Density (VCD) is obtained through
the ratio of SCD and Air Mass Factor (AMF). Among various updates, the spectral fitting no
longer considers common mode. The retrieval window (432.0 - 466.5 nm) results from a
systematic optimization and reflects trade-offs among several factors including small fitting
RMS, small fitting uncertainty, large fraction of successful retrieval and long retrieval window
length. The AMF calculation uses the latest OMI $O_2$-$O_2$ cloud product (Veefkind et al., 2016)
and monthly variable vertical profiles from the MERRA-2 reanalysis (Gelaro et al., 2017).

The Version 4 OMI TCWV product is compared against the GPS network data over land

and the SSMIS microwave observations over the oceans for 2006. Version 4 OMI TCWV has
much smaller bias than Version 3 and will replace previous versions on the AVDC website.
Version 4 OMI TCWV is characterized under difference cloud conditions. Under "clear sky"
condition (cloud fraction < 5% and cloud top pressure > 750 mb), the mean of OMI-GPS over
land is 0.85 mm with a standard deviation of 5.2 mm, and the best agreement (mean difference =
0.2 mm) occurs when TCWV is between 10 mm and 20 mm; the mean of OMI-SSMIS over the
oceans is 1.2 - 1.9 mm with a standard deviation of 6.5 - 6.8 mm, and the best agreement (mean
difference = 0.3 - 1.5 mm) occurs when TCWV is between 20 mm and 30 mm. The correlation
coefficient between OMI TCWV and the reference datasets realizes the largest gain when the
cloud fraction threshold is increased from 5% to 15%, but the bias and standard deviation also
become larger. Larger cloud fraction thresholds lead to larger biases and scatters without
improving the correlation coefficients. Thus, we recommend filtering OMI data with cloud
fraction < 5% to 15% and cloud top pressure > 750 mb, in addition to main data quality flag = 0
and fitting RMS < 0.005. Relaxing the cloud top pressure threshold (e.g., from p > 750 mb to p >
300 mb) has a similar effect as relaxing the cloud fraction threshold (e.g., from f < 5% to f <

15%).

As example applications of the Version 4 OMI TCWV data across a variety of temporal and

spatial scales, this paper examines the climate pattern associated with El Niño / La Niña, the



enhanced humidity during a week-long corn sweat event in the Midwest US, and the linear band
of high TCWV associated with an intense Atmospheric River which made landfall on the west
coast of North America. Strong signals are found in OMI TCWV for all three examples. A data
assimilation experiment shows that the OMI TCWV data can help improve WRF's skill of
simulating the shape and intensity of the AR, as well as the accumulated rainfall near the coast.
Futher improvement of the product can proceed from both spectral fitting and AMF
calculation, such as, instrument slit-function and solar irradiance for spectral fitting, aerosol
correction and surface bi-directional reflectance for AMF calculation.

**Data availability**
The GPS network data are downloaded from NCAR (rda.ucar.edu/datasets/ds721.1). The SSMIS
data used in this paper are downloaded from the Remote Sensing Systems
(http://www.remss.com/support/data-shortcut/). The Multivariate ENSO Indices are downloaded
from NOAA (https://www.esrl.noaa.gov/psd/enso/mei/table.html). OMI TCWV and ozone
profile data are released through the Aura Validation Data Center (https://avdc.gsfc.nasa.gov/).

**Author contribution**
Huiqun Wang optimized the OMI TCWV retrieval algorithm, performed the data validation
and tested most of the data application described in this paper. Amir Souri performed the WRF
simulation and data assimilation experiment presented in this paper. Gonzalo Gonzalez Abad
developed and maintained the SAO retrieval code and implemented OMI TCWV data production
at the Aura Validation Data Center. Xiong Liu developed the OMI ozone profile retrieval and
provided the relevant data used in the AR application. Kelly Chance is the PI of the NASA grant,
and is responsible for the overall direction and execution of the project. Huiqun Wang prepared
the manuscript with contributions from all co-authors. All authors contributed to technical and
scientific discussions during this project.

**Competing interests**



The authors declare that they have no conflict of interest.

**Acknowledgement**
We thank NASA's ACMAP program (Grant NNX17AH47G) for support.




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
