# Peer review of "OMI Total Column Water Vapor Version 4 Validation and Applications"

_Atmospheric Measurement Techniques, 2019_

## Referee Comment (RC1) · Anonymous Referee #1 · 27 May 2019

**Review report for "OMI Total Column Water Vapour Version 4: Validation and Applications"**

*by Huiqun Wang, Amir Hossein Souri, Gonzalo Gonzalez Abad, Xiong Liu, and Kelly Chance*

**General comments**

In this manuscript, the version 4 TCWV retrieval from OMI is validated against ground-based GPS TCWV retrievals over land and SSMIS satellite microwave retrievals over land. Differences of the version 4 retrieval with previous versions have been described, although a detailed analysis of the improvement with respect to the previous version is still lacking. I will point out some specific examples where such an additional comparison might be included in the manuscript. Also the interpretation of some of the findings for the OMI TCWV differences with TCWV from GPS or SSMIS is lacking, see again below in my specific comments.

Thereafter, 3 well-chosen examples show the importance of having a global TCWV dataset, here from OMI. These are nice demonstrations of the TCWV product, but the authors might argue more what the added value of in particular OMI TCWV (and version 4) is for those applications, compared to other satellite retrievals or reanalyzes.

**Specific comments**

- Page 1, line 10: I would write out "OMI" already in the abstract, as well as WRF (on line 28).
- Page 2-3, lines 58 –60: to me, it is strange to already mention a result of the analysis in the introduction of the manuscript. I would drop this sentence.
- Page 3, lines 72-73: here again, you already mention a result of this study in the introduction. Reformulate please.
- Page 3, line 80: data filtering criteria are recommended
- Page 4, lines 96-100: rather strange formulation. I would start the sentence with "In the non-linear least square fitting, we consider…" And also, please reformulate "In addition to water vapour" to a more specific formulation as e.g. "the use of spectroscopic water vapour dataset".
- Page 4, lines 100-108: to a reader that is not entirely in the satellite data retrieval field, it might seem ought that you start the discussion here with what version 4 is not using (common mode) in the fitting. Perhaps describe first how the fitting is done with version 4 and then describe the disadvantages of the common mode.
- Page 4, lines 109-110: as it turned out that the choice of the water vapour reference spectrum really matters for the comparison between the version 3 and 4 TCWV retrievals (later in the manuscript), you might comment on why you use an "older" water vapour reference spectrum in version 4 than in version 3.
- Page 6, lines 134-139: is the compromise for the wavelength interval as retrieval window for version 4, chosen for a particular orbit number and geographical area, also tested/valid for other orbits and other areas? Please comment.

- Page 6, lines 140-145 and Fig 2.: I really do not understand what is represented in Fig 2. Is this the overall median SCD of the entire dataset or also for the same orbit and geographical area as in Fig. 1? Please specify.
- Page 8, lines 184-185: from which dataset do you obtain the "mean elevation within the corresponding 0.25°×0.25° grid square"?
- Page 9, lines 203-204: "because the fitting includes many other interference molecules whose reference spectra may also contain errors within the retrieval window" → are version 3 and version 4 not using the same reference spectra for those molecules? So the errors in those reference spectra should then give the same effect in both version 3 and 4, no?
- Page9, lines 211-212: "This indicates a positive bias of OMI against GPS for small TCWV and a negative bias for large TCWV"
- Page 11, lines 235-236: what might be the reason for the rapid increase of r from f=0.05 to f=0.15? The other parameters are changing more smoothly between the different f ranges (as well as the r for the other f ranges).
- Page 13, lines 267-268: "suggesting that OMI cloudy TCWV is larger than OMI clear TCWV in general". Come up with an explanation here.
- Page 13, lines 273-274: "In most cases, higher cloud fraction thresholds correspond to larger σ values." Give an explanation here.
- In Section 3.2, you do not compare the version 3 OMI –SSMI TCWV retrievals with the version 4 OMI – SSMI TCWV retrievals. As you did it for GPS (over land), we lack the information of the version 4 behaviour w.r.t. version 3 over the oceans.
- Page 16, lines 348-351: this part belongs to the section describing the sensitivity of the OMI-GPS TCWV differences, and not here.
- In contrast, I would add a paragraph at the end of section 3 in which you mention the overall conclusions of the OMI TCWV validation with both GPS and SSMIS (e.g. best agreement in the 10-20/30 mm range, worse for smaller & higher TCWV ranges + reasons) and some conclusions on the improvement of version 4 over version 3.
- Page 17, Fig 7a: indicate the July 2010 and July 2015 epochs on the time series of the ENSO index.
- Page 17, lines 368-373: mentioning Level 3 and Level 2 for creating the different climatologies is confusing to me. Basically, you first construct the long-term (2005-2015) July TCWV monthly mean map (climatology). Then you create the July 2010 monthly mean map, and the July 2015 monthly mean map and you calculate the differences of those monthly means with the long-term July climatology, right? Shouldn't you use exactly the same dataset (Level 2 or Level 3) for those monthly mean maps?
- Page 17, lines 374-377: personally, I would prefer not to use the verbs "increases" and "deceases" when comparing a monthly mean of a specific month with the long-term monthly mean (=anomalies), but rather reserve those verbs in describing trends in time series. I would rather use "is elevated/higher w.r.t. "
- Page 18, line 381: if you give a possible reason for the differences in details, then you should also specify what those "differences in details" are.

- Page 20, line 412: write out NARR.
- Page 20, line 418-419: Describing Figure 9, you write that "TCWV is generally lower in the run without evapotranspiration". This is true, except in the lower boundaries of the box. Where does it come from?
- Page 21, lines 439-448: You use a very detailed description of the AR event of 6-7 Nov 2006, based on datasets that are not used/shown here. Could you not describe the event shorter – process-wise – and refer to the frequently cited Neiman et al. 2008 paper for more details?
- Page 22, lines 465-466: "is consistent with the dark stripe in the upper tropospheric water vapor image obtained by GOES-11" → show similarities to the formation processes, not to datasets or observations not shown here.
- Page 24, Figure 11: please add in the figure caption that the grey color coding means no data available.
- Page 25, line 523-524: specify the "error" in the simulated AR structure (i.e. too strong southern filament of TCWV).

---

## Referee Comment (RC2) · Anonymous Referee #2 · 6 Jun 2019

In their paper, Wang et al. present an update of a total column water vapor (TCWV) retrieval in the visible spectral range using OMI spectra. They briefly document the changes made for the update and demonstrate the improvements within a validation study including measurements from microwave satellite and ground-based GPS. In addition, they show how the updated data might be used for studies on ENSO, Corn Sweat events, and atmospheric rivers.

Overall, the paper is nicely written and easy to read. However I have major concerns regarding the validation study, the drawn conclusions of this study and the case studies of possible applications. I will list my concerns below.

[Figure]

**Major concerns**

- Since this paper presents an update of an exising data set/retrieval, it is evident to clearly demonstrate that the update distinctively improves the algorithm compared to the previous versions. This is not done in this work. Hence I suggest that the authors also include comparisons between the reference data sets and the previous retrieval version.

- I am not fully convinced by the conclusions for the intercomparison between OMI data and GPS data. The linear fit has a slope of 0.82 even for clear-sky observations (radiance cloud fraction < 0.05) and for larger cloud fraction those fit results are actually missing. Additionally, I think that it is a simplification to focus on bias and standard deviation only for interpreting the data. Thus I suggest to include also the mean absolute error (MAE) in the validation study and to perform linear fits for the different cloud fraction thresholds. Furthermore, only some selected scatter plots of the intercomparison between OMI TCWV data and reference data sets are shown in this paper. I suggest that for each cloud fraction threshold the corresponding scatter plots and linear fits are displayed, which may be provided in an Appendix or Supplement to the paper.

- Although Section 4 shows very interesting insights in the application of the new data set, it does not really fit the scope of AMT and should be skipped. Nevertheless I think it might be interesting to see what the impact of other satellite data (AIRS, SSM/I, TES, etc.) will be on the respective case studies and how much better the OMI performs within this comparison. But this will be probably beyond the scope of this paper.

**General comments**

1. How strong are the cross-correlations between the considered absorption cross-sections (more precisely between H2O, liquid water, glyoxal)? Considering the retrieval fit window, since the 6n+d H2O line is partially included within this window, do you account for correction factors reported in Lampel et al. (2015)?

2. How large is the dependency on the MERRA-2 water vapor profiles and doesn't this mean that your retrieval is not fully independent from external data sets?

3. It would be very interesting to the reader to see which update step contributes most to the improvement (new fit window, new cloud product, new MERRA-2 data?).

4. Do you use the radiative or the geometric cloud fraction? The cloud fraction criterion of 0.05 seems very restrictive to me. What fraction of OMI data is filtered by this criterion?

5. Linear regression for land data: Why is the slope so bad? Please discuss in more detail the uncertainties of the SCD and the AMF.

6. El Nino study: Since you only consider clear-sky observations, your averaged data are biased. How strong is thus the influence/impact of clouds?

7. Corn Sweat study: Do you observe the increase of TCWV also in the GPS data?

8. AR study: I do not see the benefit of the description of the AR in Section 4.3.2, especially when this AR is already analysed in detail by Neimann et al. (2008). So the authors might think about skipping this section.

**Specific comments**

- line 10: please introduce the complete name for OMI (Ozone Monitoring Instrument)

- line 12: „various updates" → more specific (e.g. updated cloud product, etc.)

- line 16: geometric/radiative cloud fraction?

- line 22: I think you meant 20-30 instead of 10-30 mm

- line 27: atmospheric river

- line 37: in situ

- line 38: „ground" instead of surface

- line 41: the reference is Schröder et al. (2018)

- line 43: I would rather say clear-sky and cloudy-sky

- line 58: It is very unusual to mention results already in the introduction

- line 69: reference spectra for water vapor from the latest HITRAN database . . .

- line 74: please insert a line break

- line 86: Aura

- line 88: 1:30 PM local time (this is actually the equator crossing-time)

- line 95: The specifics of Version 4 are discussed . . .

- line 106: show

[Figure]

- line 125: „smaller toward the the lower right corner of the domain" please rephrase this

- line 131: „5x10^23 molecules/cm²" where is this value coming from?

- line 134: Shouldn't the residual be small as well as not contain any structures, i.e. it should be noisy?

- line 159: influence the AMF

- line 168 to 172: it would be very helpful to have a map showing the distribution of the GPS stations

- line 176: what are unphysical values? Please describe in more detail.

- line 180: the time window seems quite large, since water vapor can vary a lot during day. I think it would be better reduce the time range to plus/minus 1.5 hours

- line 185: which elevation database are you using?

- line 186: „we consider the OMI and GPS data that are less than 75 mm." why not higher values (e.g. 100 mm)?

- line 195 and 196: the references have to be switched

- line 209: please provide $R^2$ of the linear fit

- line 279: Isn't this also an instrumental issue (poor signal to noise ratio of OMI?)

- line 290: cycles

- line 291: which special cases?

- line 293 to 295: now you are using different requirements for the fit (e.g. higher TCWV threshold).

- line 373: Figure 7cd → Figure 7c) and 7d)

- line 376 and 377: decrease

- line 492: what is the weight of the OMI observations for your assimilated data? Can you provide a map for that? • line 556: „we recommend to consider only OMI data ...“

- Table 3: Please indicate the fraction of used data points to available data points in percent. Also split up the regression column into slope and intercept and coefficient of determination ($R^2$).

- Figure 1: please remove panel d) and replace it with Figure 2

- Figure 3: Please include a comparison for Version 3 and the linear fits in the scatter plots. Also colorbar in the bottom panel has no labels.

- Figure 10: Please zoom into the region of interest.

- Figure 12: Why does the model simulate rainfall in the northwest of Oregon even for the case with assimilated OMI data?

**References**

- Lampel, J., et al. "On the relative absorption strengths of water vapour in the blue wavelength range." Atmospheric Measurement Techniques 8.10 (2015): 4329-4346.

- Neiman, Paul J., et al. "Meteorological characteristics and overland precipitation impacts of atmospheric rivers affecting the West Coast of North America based on eight years of SSM/I satellite observations." Journal of Hydrometeorology 9.1 (2008): 22-47.

- Schröder, Marc, et al. "The GEWEX Water Vapor Assessment archive of water vapour products from satellite observations and reanalyses." Earth System Science Data 10.2 (2018): 1093-1117.

---

## Author Comment (AC1) · 23 Jul 2019

**Response to Review report for "OMI Total Column Water Vapour Version 4: Validation and Applications"**

*by Huiqun Wang, Amir Hossein Souri, Gonzalo Gonzalez Abad, Xiong Liu, and Kelly Chance*

General comments

In this manuscript, the version 4 TCWV retrieval from OMI is validated against ground-based GPS TCWV retrievals over land and SSMIS satellite microwave retrievals over land. Differences of the version 4 retrieval with previous versions have been described, although a detailed analysis of the improvement with respect to the previous version is still lacking. I will point out some specific examples where such an additional comparison might be included in the manuscript. Also the interpretation of some of the findings for the OMI TCWV differences with TCWV from GPS or SSMIS is lacking, see again below in my specific comments. Thereafter, 3 well-chosen examples show the importance of having a global TCWV dataset, here from OMI. These are nice demonstrations of the TCWV product, but the authors might argue more what the added value of in particular OMI TCWV (and version 4) is for those applications, compared to other satellite retrievals or reanalyzes.

Thank you very much for the thorough and constructive review. We have improved the manuscript following your suggestions. The example applications are intended to test and show that there is value in the OMI TCWV dataset, and therefore, the data can contribute to the overall understanding of water vapor. Comparisons with other satellite datasets or reanalyzes for the added value of OMI TCWV are left for future work.

Please find our detailed response below.

**Specific comments**

☐ Page 1, line 10: I would write out "OMI" already in the abstract, as well as WRF (on line 28).

We have now written them out.

☐ Page 2-3, lines 58 –60: to me, it is strange to already mention a result of the analysis in the introduction of the manuscript. I would drop this sentence.

The sentence has been deleted.

☐ Page 3, lines 72-73: here again, you already mention a result of this study in the introduction. Reformulate please.

The sentence has been deleted.

☐ Page 3, line 80: data filtering criteria are recommended

"is" has been changed to "are".

☐ Page 4, lines 96-100: rather strange formulation. I would start the sentence with "In the non-linear least square fitting, we consider…" And also, please reformulate "In addition to water vapour" to a more specific formulation as e.g. "the use of spectroscopic water vapour dataset".

The sentence has been rephrased following the suggestion.

☐ Page 4, lines 100-108: to a reader that is not entirely in the satellite data retrieval field, it might seem ought that you start the discussion here with what version 4 is not using (common mode) in

the fitting. Perhaps describe first how the fitting is done with version 4 and then describe the disadvantages of the common mode.

The elements considered in the Version 4.0 nonlinear least square fitting are explained in the previous sentence. The intention of this sentence is to point out the difference with previous versions. For readers who are unfamiliar with common mode, we have added the reference González Abad et al. (2015).

☐ Page 4, lines 109-110: as it turned out that the choice of the water vapour reference spectrum really matters for the comparison between the version 3 and 4 TCWV retrievals (later in the manuscript), you might comment on why you use an "older" water vapour reference spectrum in version 4 than in version 3.

We have added a couple of sentences to explain the rationale. It is primarily driven by the validation results. In addition, through personal communication with the HITRAN group at the Smithsonian Astrophysical Observatory, we have recently learned that HITRAN 2016 has some issues with water vapor in the blue wavelength range and that spectroscopic improvements are being made for the next HITRAN release.

☐ Page 6, lines 134-139: is the compromise for the wavelength interval as retrieval window for version 4, chosen for a particular orbit number and geographical area, also tested/valid for other orbits and other areas? Please comment.

We have changed "we use OMI Orbit number 10426 … as an example to …" to "we randomly selected OMI orbit number 10426 to…". We tested the result with Orbit 10423 (which cut across the Pacific near the dateline). The patterns exhibited by the variables are similar, though the values for SCD and SCD uncertainties are slightly higher, as Orbit 10423 is over the ocean.

☐ Page 6, lines 140-145 and Fig 2.: I really do not understand what is represented in Fig 2. Is this the overall median SCD of the entire dataset or also for the same orbit and geographical area as in Fig. 1? Please specify.

Following the other reviewer's suggestion, we have combined the original Figure 1 and Figure 2 into one figure. In the figure caption, we have added "for OMI Orbit number 10426".

☐ Page 8, lines 184-185: from which dataset do you obtain the "mean elevation within the corresponding 0.25°×0.25° grid square"?

The dataset was downloaded from www.temis.nl/data/topo/dem2grid.html in December 2015. The ultimate data source is USGS. A comment about this has been added.

☐ Page 9, lines 203-204: "because the fitting includes many other interference molecules whose reference spectra may also contain errors within the retrieval window" ☐ are version 3 and version 4 not using the same reference spectra for those molecules? So the errors in those reference spectra should then give the same effect in both version 3 and 4, no?

This sentence has been deleted. Version 3 does not include the Vibrational Raman Scattering of air, but Version 4 does. We have recently found, through personal communication with the HITRAN group, that the HITRAN 2016 water vapor spectrum in the blue wavelength range is adversely affected by a line broadening issue. It is therefore not surprising that HITRAN 2008 can lead to lower bias than HITRAN 2016.

☐ Page9, lines 211-212: "This indicates a positive bias of OMI against GPS for small TCWV and a negative bias for large TCWV"

The sentence has been changed following the advice.

☐ Page 11, lines 235-236: what might be the reason for the rapid increase of r from f=0.05 to f=0.15? The other parameters are changing more smoothly between the different f ranges (as well as the r for the other f ranges).

Firstly, the error in cloud top pressure decreases with cloud fraction (Veefkind et al., 2016). As a result, f = 0.05 corresponds to the largest uncertainty in cloud top pressure and the error will propagate to OMI TCWV through AMF, leading to smaller correlation coefficient. Secondly, this is related to the effective dynamical range of TCWV. There is a larger fraction of data pairs with TCWV > 40 mm for f = 0.15 than for f = 0.05. A larger dynamical range generally favors a larger correlation coefficient. The explanation has been added.

☐ Page 13, lines 267-268: "suggesting that OMI cloudy TCWV is larger than OMI clear TCWV in general". Come up with an explanation here.

We have added a sentence to explain. Basically, other things being equal, cloud formation indicates water vapor saturation and therefore higher TCWV than that under clear-sky condition.

☐ Page 13, lines 273-274: "In most cases, higher cloud fraction thresholds correspond to larger σ values." Give an explanation here.

This is consistent with the larger dynamical range (due to a larger fraction of data with high TCWV) for larger cloud fraction thresholds. The relative scatter, however, shows little dependence on cloud fraction threshold. A comment about this has been added.

☐ In Section 3.2, you do not compare the version 3 OMI –SSMI TCWV retrievals with the version 4 OMI – SSMI TCWV retrievals. As you did it for GPS (over land), we lack the information of the version 4 behaviour w.r.t. version 3 over the oceans.

We have added the information. Essentially, Version 3 OMI TCWV has significantly larger bias than Version 4.

☐ Page 16, lines 348-351: this part belongs to the section describing the sensitivity of the OMI-GPS TCWV differences, and not here.

We mis-typed OMI-SSMIS as OMI-GPS. The error has been corrected. Thanks for catching it.

☐ In contrast, I would add a paragraph at the end of section 3 in which you mention the overall conclusions of the OMI TCWV validation with both GPS and SSMIS (e.g. best agreement in the 10-20/30 mm range, worse for smaller & higher TCWV ranges + reasons) and some conclusions on the improvement of version 4 over version 3.

The overall conclusions from the comparisons are summarized in the "Summary and Conclusion" section.

☐ Page 17, Fig 7a: indicate the July 2010 and July 2015 epochs on the time series of the ENSO index.

We have drawn dashed vertical lines to indicate the epochs in the plot.

☐ Page 17, lines 368-373: mentioning Level 3 and Level 2 for creating the different climatologies is confusing to me. Basically, you first construct the long-term (2005-2015) July TCWV monthly mean map (climatology). Then you create the July 2010 monthly mean map, and the July 2015 monthly mean map and you calculate the differences of those monthly means with the long-term July climatology, right? Shouldn't you use exactly the same dataset (Level 2 or Level 3) for those monthly mean maps?

The procedure described above is indeed what we used for the figure. Averaging the monthly Level 3 July data is an alternative way of composing the July climatology. It does not make any noticeable difference for the purpose of this figure.

☐ Page 17, lines 374-377: personally, I would prefer not to use the verbs "increases" and "deceases" when comparing a monthly mean of a specific month with the long-term monthly mean (=anomalies), but rather reserve those verbs in describing trends in time series. I would rather use "is elevated/higher w.r.t. "

We have changed to "higher/lower".

☐ Page 18, line 381: if you give a possible reason for the differences in details, then you should also specify what those "differences in details" are.

We have deleted this part, as it is not essential for this paper. Readers who are interested in the details can compare with Shi et al. (2018).

☐ Page 20, line 412: write out NARR.

It has been written out.

☐ Page 20, line 418-419: Describing Figure 9, you write that "TCWV is generally lower in the run without evapotranspiration". This is true, except in the lower boundaries of the box. Where does it come from?

The higher TCWV in the No ET run near the southern boundary reflects the non-linear water vapor transport from the Gulf region. Note, turning off evapotranspiration not only affects the water vapor flux from the surface, but also influences other meteorological variables, such as temperature and winds. Thus, there is a difference in the water vapor flux across the domain boundary. A comment has been added in the paper.

☐ Page 21, lines 439-448: You use a very detailed description of the AR event of 6-7 Nov 2006, based on datasets that are not used/shown here. Could you not describe the event shorter – process-wise – and refer to the frequently cited Neiman et al. 2008 paper for more details?

We have shortened the description and combined the original Section 4.3.1 and 4.3.2 into one subsection.

☐ Page 22, lines 465-466: "is consistent with the dark stripe in the upper tropospheric water vapor image obtained by GOES-11" ☐show similarities to the formation processes, not to datasets or observations not shown here.

We have deleted this part and pointed out that the feature is associated with the same extra-tropical cyclone as the AR is.

☐ Page 24, Figure 11: please add in the figure caption that the grey color coding means no data available.

We have added in the figure caption "Gray color indicates area with no SSM/I data".

☐ Page 25, line 523-524: specify the "error" in the simulated AR structure (i.e. too strong southern filament of TCWV).

We have specified the error according to the suggestion.

---

## Author Comment (AC2) · 23 Jul 2019

In their paper, Wang et al. present an update of a total column water vapor (TCWV) retrieval in the visible spectral range using OMI spectra. They briefly document the changes made for the update and demonstrate the improvements within a validation study including measurements from microwave satellite and ground-based GPS. In addition, they show how the updated data might be used for studies on ENSO, Corn Sweat events, and atmospheric rivers. Overall, the paper is nicely written and easy to read. However I have major concerns regarding the validation study, the drawn conclusions of this study and the case studies of possible applications. I will list my concerns below.

Thanks for the thorough and constructive review. We have revised the paper accordingly. Please find our response to each point below.

**Major concerns**
• Since this paper presents an update of an exising data set/retrieval, it is evident to clearly demonstrate that the update distinctively improves the algorithm compared to the previous versions. This is not done in this work. Hence I suggest that the authors also include comparisons between the reference data sets and the previous retrieval version.

We have added information for the comparisons between Version 3.0 OMI data and reference TCWV datasets. Essentially, Version 3.0 OMI data have significantly larger bias than Version 4.0.

• I am not fully convinced by the conclusions for the intercomparison between OMI data and GPS data. The linear fit has a slope of 0.82 even for clear-sky observations (radiance cloud fraction < 0.05) and for larger cloud fraction those fit results are actually missing. Additionally, I think that it is a simplification to focus on bias and standard deviation only for interpreting the data. Thus I suggest to include also the mean absolute error (MAE) in the validation study and to perform linear fits for the different cloud fraction thresholds. Furthermore, only some selected scatter plots of the intercomparison between OMI TCWV data and reference data sets are shown in this paper. I suggest that for each cloud fraction threshold the corresponding scatter plots and linear fits are displayed, which may be provided in an Appendix or Supplement to the paper.

Following the suggestion, we have added OMI versus GPS scatter plots for different cloud fraction thresholds. The correlation coefficient and regression line are actually better for cloud fraction f ≥ 0.15 than for f = 0.05. The best regression line has a slope close to one. The linear regression is worse for f = 0.05 because of (a) larger error in cloud top pressure in OMCLDO2 product for smaller cloud fraction (b) smaller effective dynamical range of TCWV for smaller cloud fraction threshold. Information for the mean absolute error has been added. The MAE is smaller than the standard deviation for the comparisons.

• Although Section 4 shows very interesting insights in the application of the new data set, it does not really fit the scope of AMT and should be skipped. Nevertheless, I think it might be

interesting to see what the impact of other satellite data (AIRS, SSM/I, TES, etc.) will be on the respective case studies and how much better the OMI performs within this comparison. But this will be probably beyond the scope of this paper.

The example applications are intended to show that there is value in the OMI TCWV dataset. In a sense, the applications also serve as an "indirect" validation of the dataset, as a useful dataset is expected to contain well-known signals. As different satellites have different overpass time, resolution and limitation, we believe that each satellite product contributes to the overall understanding of water vapor in its unique way. Comparisons with other satellite datasets for the added value of OMI TCWV are indeed beyond the scope of this paper.

**General comments**
1. How strong are the cross-correlations between the considered absorption crosssections (more precisely between H2O, liquid water, glyoxal)? Considering the retrieval fit window, since the 6n+d H2O line is partially included within this window, do you account for correction factors reported in Lampel et al. (2015)?

The linear correlation coefficient between H2O and glyoxal is 0.009, and that between H2O and liquid water is -0.20. The water vapor reference spectrum used in Version 4.0 is based on the original HITRAN 2008 which does not consider the correction in Lampel et al. (2015). A comment about this has been added. The water vapor spectrum in the blue wavelength range is being improved by the HITRAN group. We expect that the next HITRAN release will be better.

2. How large is the dependency on the MERRA-2 water vapor profiles and doesn't this mean that your retrieval is not fully independent from external data sets?

We compared the TCWV computed using the MERRA-2 profiles with those computed using the ERA-Interim profiles for July 2006. The result shows that the standard deviation of the difference can be significant (~ 3mm). The information has been added to the text. To mitigate the dependence of TCWV on external datasets, scattering weights are provided in the Level 2 OMI product. Users can convolve the scattering weights with the profiles of their choice to calculate AMF and adjust TCWV.

3. It would be very interesting to the reader to see which update step contributes most to the improvement (new fit window, new cloud product, new MERRA-2 data?).

Between Version 3.0 and 4.0, the reference water vapor spectrum leads to the largest difference. This is mentioned when we discuss supplementary Figure 1.

4. Do you use the radiative or the geometric cloud fraction? The cloud fraction criterion of 0.05 seems very restrictive to me. What fraction of OMI data is filtered by this criterion?

We used the cloud fraction reported in the OMCLDO2 product (Veefkind et al., 2016). We have clarified this in the revised paper. In Veefkind et al. (2016), this effective cloud fraction is calculated using the reflectance at the top-of-atmosphere, for the clear part and for the cloudy part, and can be considered as a radiative cloud fraction.

On a typical day (July 1, 2016), among the OMI data that pass the MDQFL and TCWV range test, f<0.05 accounts for about 35% of the data. The information has been added.

5. Linear regression for land data: Why is the slope so bad? Please discuss in more detail the uncertainties of the SCD and the AMF.

This is related to the relatively large bias for TCWV < 10 mm. The slope improves if these data pairs are excluded from the linear regression. A comment about this has been added. Moreover, the slope is also due to the smaller dynamical range of TCWV for cloud fraction < 0.05. For larger cloud fraction thresholds, there is a larger fraction of data pairs with higher TCWV values, and the regression slopes improve. The best regression line has a slope close to one (for cloud fraction < 0.25 or 0.35). A figure has been added to show this.

Typical uncertainties of SCD can be found in Figure 1 and supplementary Figure 1. For the uncertainty related to gas profiles for AMF, please see our reply to (2). For the uncertainty related to scattering weights for AMF, we have conducted error propagation analysis for a typical orbit, results show that most AMF error is <3%, though it can be up to 15% for cloudy pixels over land. The information has been added in Section 2.

6. El Nino study: Since you only consider clear-sky observations, your averaged data are biased. How strong is thus the influence/impact of clouds?

For the El Niño study, we used cloud fraction < 0.15 and cloud top pressure > 750 hPa to filter OMI data (in addition to other usual criteria). This choice is based on the validation results presented previously. Using stricter criteria for clouds will result in lots more missing data in the map, using less restrictive criteria for clouds will incur larger data bias which will be hard to disentangle from the signal. Thus, the influence of clouds on the pattern is not discussed in this paper.

7. Corn Sweat study: Do you observe the increase of TCWV also in the GPS data?

Yes. Several GPS stations over the area observed an increase of TCWV during the event, though coincident OMI data at the particular stations are not found. Supplementary figure has been added.

8. AR study: I do not see the benefit of the description of the AR in Section 4.3.2, especially when this AR is already analysed in detail by Neimann et al. (2008). So the authors might think about skipping this section.

We have shortened the description of this event and merged the original Section 4.3.1 and 4.3.2 into one subsection.

**Specific comments**
• line 10: please introduce the complete name for OMI (Ozone Monitoring Instrument)
  We have added the complete name.
• line 12: „various updates" _ more specific (e.g. updated cloud product, etc.)

We have changed "various updates" to "reference spectra and gas profiles".

• line 16: geometric/radiative cloud fraction?

It is the effective cloud fraction reported in the OMCLDO2 product (Veefkind et al., 2016). It is based on radiances and therefore can be considered as a radiative cloud fraction. A sentence about this has been added to clarify.

• line 22: I think you meant 20-30 instead of 10-30 mm

A change has been made to summarize the result more accurately.

• line 27: atmospheric river

It has been changed following the suggestion.

• line 37: in situ

It has been changed following the suggestion.

• line 38: „ground" instead of surface

It has been changed following the suggestion.

• line 41: the reference is Schröder et al. (2018)

It has been corrected.

• line 43: I would rather say clear-sky and cloudy-sky

It has been changed following the suggestion.

• line 58: It is very unusual to mention results already in the introduction

The sentence has been deleted.

• line 69: reference spectra for water vapor from the latest HITRAN database . . .

It has been changed following the suggestion.

• line 74: please insert a line break

A line break has been inserted.

• line 86: Aura

It has been changed following the suggestion.

• line 88: 1:30 PM local time (this is actually the equator crossing-time)

It has been changed to "1:30 PM equator crossing time".

• line 95: The specifics of Version 4 are discussed . . .

It has been changed following the suggestion.

• line 106: show

It has been changed following the suggestion.

• line 125: „smaller toward the the lower right corner of the domain" please rephrase this

The sentence has been rephrased to be more specific about what the lower right corner of the domain means.

• line 131: „5x10^23 molecules/cm²" where is this value coming from?

The threshold corresponds to a SCD of about 149.45 mm. It is meant to filter out large outliers. For reference, the largest TCWV of the GPS and SSMIS datasets (Section 3) is about 75 mm. At low latitudes where TCWV is high, more than 90% of the AMFs are between 0.5 and 2.0. We have added the information to the text.

• line 134: Shouldn't the residual be small as well as not contain any structures, i.e. it should be noisy?

We have changed "reduce the residual" to "reduce the residual's amplitude and structure".

• line 159: influence the AMF

It has been changed following the suggestion.

• line 168 to 172: it would be very helpful to have a map showing the distribution of the GPS stations.

We have added (Wang et al., 2016) as a reference for the distribution of the stations on a map.

• line 176: what are unphysical values? Please describe in more detail.
  We have changed it to "negative or extremely large (TCWV > 75 mm) values".
• line 180: the time window seems quite large, since water vapor can vary a lot during day. I think it would be better reduce the time range to plus/minus 1.5 hours.
  We have changed the time window to 1200 LT - 1500 LT.
• line 185: which elevation database are you using?
  The 0.25°×0.25° topography was downloaded from www.temis.nl/data/topo/dem2grid.html.
• line 186: „we consider the OMI and GPS data that are less than 75 mm." why not higher values (e.g. 100 mm)?
  The largest TCWV of the GPS data used is about 75 mm. A comment about this has been added.
• line 195 and 196: the references have to be switched
  The references have been switched. Thanks for catching that.
• line 209: please provide R² of the linear fit
  It has been provided.
• line 279: Isn't this also an instrumental issue (poor signal to noise ratio of OMI?)
  We have rephrased in term of "low signal-to-noise ratio when TCWV < 10 mm in the OMI retrieval".
• line 290: cycles
  It has been deleted.
• line 291: which special cases?
  This part of the sentence has been deleted. Users who are interested in the details can refer to Diedrich et al. (2016).
• line 293 to 295: now you are using different requirements for the fit (e.g. higher TCWV threshold).
  We have changed the data filtering criteria so that they are consistent with the ones used before. The corresponding figures and discussions have also been updated as needed.
• line 373: Figure 7cd _ Figure 7c) and 7d)
  We have made the change.
• line 376 and 377: decrease
  Following the other reviewer's suggestion, we have changed increase/decrease to higher/lower.
• line 492: what is the weight of the OMI observations for your assimilated data? Can you provide a map for that?
  The weight varies for each simulation window depending on the data quality and data density. A single map cannot describe the process, therefore, it is not provided in the paper.
• line 556: „we recommend to consider only OMI data ..."
The change has been made.
• Table 3: Please indicate the fraction of used data points to available data points in percent. Also split up the regression column into slope and intercept and coefficient of determination (R²).
We have made the changes.
• Figure 1: please remove panel d) and replace it with Figure 2
  We have made the change.
• Figure 3: Please include a comparison for Version 3 and the linear fits in the scatter plots. Also colorbar in the bottom panel has no labels.

We have fixed the color bar. We have included the comparisons for Version 3 OMI in the text. Given the focus and length of this paper, we feel that it is not essential to include scatter plots for Version 3.

• Figure 10: Please zoom into the region of interest.

We have made the change.

• Figure 12: Why does the model simulate rainfall in the northwest of Oregon even for the case with assimilated OMI data?

Admittedly, even with data assimilation, the model is still not perfect. Errors in both the model and the data, as well as the amount and distribution of the data, contribute to the error in the assimilation result. For the example in Figure 12, we are glad to see that the model does a better job within the red box when OMI data are used. A detailed investigation of the assimilation error is beyond the scope of this paper. A comment about this has been added.

**References**

• Lampel, J., et al. "On the relative absorption strengths of water vapour in the blue wavelength range." Atmospheric Measurement Techniques 8.10 (2015): 4329-4346.
• Neiman, Paul J., et al. "Meteorological characteristics and overland precipitation impacts of atmospheric rivers affecting the West Coast of North America based on eight years of SSM/I satellite observations." Journal of Hydrometeorology 9.1 (2008): 22-47.
• Schröder, Marc, et al. "The GEWEX Water Vapor Assessment archive of water vapour products from satellite observations and reanalyses." Earth System Science Data 10.2 (2018): 1093-1117.

Thanks for providing the references. We have added them in the paper.

[Figure]

**Figure 1.** Sensitivity of the retrieval to the start and end wavelengths (nm) of the retrieval window for OMI Orbit number 10426. (a) Median of fitting RMS×10$^4$; (b) median of water vapor SCD fitting uncertainty in mm; (c) valid fraction for retrievals; (d) median SCD in mm.

[Figure]

Figure 4. Normalized joint distributions of GPS versus Version 4.0 OMI TCWV for different cloud fraction thresholds. Results are derived from the co-located data pairs for 2006. The OMI data filtering criteria are the same as those for Figure 3. In each panel, the 1:1 line is plotted in black, the linear regression line is plotted in gray and indicated by the formula in the lower right corner.

| f | N | P (%) | Mean | σ | MAE | r | $R^2$ | b | k |
|---|---|---|---|---|---|---|---|---|---|
| 0.05 | 1,048,879 | 7.4 | 0.02 | 7.11 | 5.39 | 0.82 | 0.67 | 1.43 | 0.95 |
| 0.15 | 2,837,032 | 20.0 | 1.38 | 7.82 | 5.84 | 0.84 | 0.71 | 0.70 | 1.02 |
| 0.25 | 3,932,468 | 27.8 | 2.20 | 8.09 | 6.09 | 0.84 | 0.71 | 1.11 | 1.04 |
| 0.35 | 4,819,185 | 34.0 | 2.73 | 8.22 | 6.24 | 0.85 | 0.72 | 1.45 | 1.05 |
| 0.45 | 5,537,003 | 39.1 | 3.07 | 8.26 | 6.32 | 0.85 | 0.72 | 1.62 | 1.06 |

**Table 3.** Effect of cloud fraction threshold on the comparison between SSMIS and Version 4.0 OMI TCWV for July 2006. f: OMI cloud fraction threshold; N: number of qualifying data pairs; P: Percentage of qualifying data pairs with respect to the total number of qualifying SSMIS data points; Mean: mean of OMI-SSMIS in mm; σ: standard deviation of OMI-SSMIS in mm; MAE: Mean absolute error |OMI-SSMIS| in mm; r: correlation coefficient between SSMIS and OMI; $R^2$: coefficient of determination for linear regression OMI = b + k * SSMIS, where OMI and SSMIS are in mm; b: Intercept of linear regression; k: slope of linear regression.

[Figure]

**Figure 10.** The Level 3 (top row) climatology, (middle row) data on November 6th, 2016 and (bottom row) anomaly on November 6th, 2016 with respect to the climatology for (left column) Version 4.0 OMI TCWV (mm, 0.5°×0.5°) and (right column) OMI ozone mixing ratio (ppb, 1°×1°) interpolated to 200 mb.

[Figure]

Supplementary Figure 1. Black curve shows the median relative SCD uncertainty for each 10 mm SCD bin (left axis). Blue curve shows the fraction of data points that fall within each 10 mm SCD bin (right axis). Results are derived from OMI orbit number 10426.

[Figure]

Supplementary Figure 2. (a) Version 4.0 versus Version 3.0 AMF comparison; (b) Version 4.0 versus Version 3.0 SCD comparison; (c) Version 4.0 versus Test 1 SCD comparison. Test 1 has the same setting as Version 4.0 except that water vapor reference spectrum is from HITRAN 2016. All results are for OMI orbit number 10423.

[Figure]

Supplementary Figure 3. Time series of TCWV (mm) observed by each GPS station indicated in the top panel for July 2016. The horizontal dashed lines indicate the mean TCWV for July. The two dotted vertical lines bracket the corn sweat time period discussed in the paper.

[Figure]

Supplementary Figure 4. Time series of surface pressure (hPa) observed by each GPS station indicated in the top panel for July 2016. The horizontal dashed lines indicate the mean surface pressure for July. The two dotted vertical lines bracket the corn sweat time period discussed in the paper.

---

## Author Response (AR2)

Dear authors,
thank you for taking all my remarks into account. I'm left with two minor issues, both on page 18.
* line 383: you always considered the year 2006 in your comparisons, and January and July 2006 in particular in Section 3.2. So, why are you comparing OMI version 3.0 with SSMIS for July 2007 here? Or is it a typo?
* lines 390-391: Why this inconsistency in filtering out pixels affected by rain between Fig. 6 and Table 3? Please give a reason for this approach.

Dear reviewer,

Thanks for your review.

For line 383, it is a typo. The year used should be 2006. Thanks for catching it. We have made the correction.

For line 390-391, SSMIS data have lower quality when there is precipitation. Thus, we have filtered out pixels with rain in both Figure 6 and Table 3. However, Figure 6 is dedicated to the comparison under "clear-sky" condition and Table 3 is for investigating the influence of different cloud fractions. We have therefore filtered out SSMIS pixels with cloud liquid water in Figure 6 and kept those pixels in Table 3. A comment about this has been added to the paper near line 391.

Regards, from all authors

[revised manuscript text omitted]